

# Ensemble analysis and forecast of ecosystem indicators in the North Atlantic using ocean colour observations and prior statistics from a stochastic NEMO/PISCES simulator

Mikhail Popov[1], Jean-Michel Brankart[1], Arthur Capet[1,2], Emmanuel Cosme[1], and Pierre Brasseur[1]

[1]Univ. Grenoble Alpes, CNRS, IRD, Grenoble INP, IGE, Grenoble, France
[2]Operational Directorate Natural Environment, Royal Belgian Institute of Natural Sciences, Brussels, Belgium

**Correspondence:** Jean-Michel Brankart (jean-michel.brankart@univ-grenoble-alpes.fr)

**Abstract.** This study is anchored in the H2020 SEAMLESS project (www.seamlessproject.org), which aims to develop ensemble assimilation methods to be implemented in Copernicus Marine Service monitoring and forecasting systems, in order to operationally estimate a set of targeted ecosystem indicators in various regions, including uncertainty estimates. In this paper, a simplified approach is introduced to perform a 4D (space-time) ensemble analysis describing the evolution of the
ocean ecosystem. An example application is provided, which covers a limited time period in a limited subregion of the North Atlantic (between 31°W and 21°W, between 44°N and 50.5°N, between March 15 and June 15, 2019, at a 1/4° and a 1 day resolution). The ensemble analysis is based on prior ensemble statistics from a stochastic NEMO/PISCES simulator. Ocean colour observations are used as constraints to condition the 4D prior probability distribution.

As compared to classic data assimilation, the simplification comes from the decoupling between the forward simulation using
the complex modelling system and the update of the 4D ensemble to account for the observation constraint. The shortcomings and possible advantages of this approach for biogeochemical applications are discussed in the paper. The results show that it is possible to produce a multivariate ensemble analysis continuous in time and consistent with the observations. Furthermore, we study how the method can be used to extrapolate analyses calculated from past observations into the future. The resulting 4D ensemble statistical forecast is shown to contain valuable information about the evolution of the ecosystem for a few days
after the last observation. However, as a result of the short decorrelation time scale in the prior ensemble, the spread of the ensemble forecast is increasing quickly with time. Throughout the paper, a special emphasis is given to discussing the statistical reliability of the solution.

Two different methods have been applied to perform this 4D statistical analysis and forecast: the analysis step of the Ensemble Transform Kalman Filter (with domain localization) and a Monte Carlo Markov Chain (MCMC) sampler (with covariance
localization), both enhanced by the application of anamorphosis to the original variables. Despite being very different, the two algorithms produce very similar results, thus somehow validating each other. As shown in the paper, the decoupling of the statistical analysis from the dynamical model allows us to restrict the analysis to a few selected variables and, at the same time, to produce estimates of additional ecological indicators (in our example: phenology, trophic efficiency, downward flux of



particulate organic matter). This approach can easily be appended to existing operational systems to focus on dedicated users'
requirements, at small additional cost, as long as a reliable prior ensemble simulation is available. It can also serve as a baseline
to compare with the dynamical ensemble forecast, and as a possible substitute whenever useful.

# 1  Introduction

Combining numerical models with observational data to reconstruct the past evolution of ocean biogeochemistry and to predict
its future evolution has been a major objective of operational ocean forecasting centres for many years, motivated both by the
marine user's needs and by advances in scientific knowledge of the ocean functioning (Gehlen et al., 2015).

In order to support decision-making or faithful scientific assessment, the marine ecosystem and biogeochemical parameters
to be estimated (e.g. phenology, trophic efficiency, downward carbon flux, etc.) must be supplemented with uncertainties to
quantify the robustness of the information produced and quantify the likelihood of estimates (Modi et al., 2022). This motivates
the development of systems capable of producing information which is probabilistic in nature. This study is anchored in the
H2020 SEAMLESS project, which aims to develop ensemble assimilation methods to be implemented in Copernicus Marine
Service monitoring and forecasting systems, in order to operationally estimate a set of targeted ecosystem indicators in the
different regions covered, including uncertainty estimates.

Despite their cost with high-dimensional systems, ensemble methods based on Monte Carlo simulations are well suited to
generate samples of the probability distributions of the quantities of interest, as is already implemented today by some teams of
the OceanPredict program (Fennel et al., 2019). The standard approach relies on implementations of sequential ensemble data
assimilation methods that typically consist of variants of the Ensemble Kalman filter (Evensen, 2007). The data assimilation
algorithms typically perform ensemble analysis and predictions in sequence to integrate observational information such as
satellite ocean color and ARGO BGC profile data into coupled 3D physical/biogeochemical models (Gutknecht et al., 2022).

In these ensemble data assimilation systems, the most expensive numerical component is the coupled physical/biogeochem-
ical model that is used to perform the ensemble simulations, because it is usually sought to run at high horizontal resolution
and because a few dozens of members are usually necessary to obtain ensemble statistics that are accurate enough for data
assimilation. The aim of this paper is to demonstrate that it is possible to make an additional use of these expensive data, ob-
tained from a prior ensemble model simulation (not yet conditioned on observations), to produce a statistically-driven analysis
and forecast for selected key model variables. A secondary aim of this paper is to illustrate the extension of the approach to
estimate ecological indicators diagnosed from model state variables.

For instance, using the statistics of the ensemble, it is not difficult, at least in principle, to obtain a 4D multivariate statistical
analysis based on all available observations over a given time period (possibly quite long, typically between 1 month and 1 year,
maybe longer). This just corresponds to applying any standard Bayesian observational update algorithm in four dimensions
(4D) to condition the prior ensemble on the observations, and thus produce the corresponding 4D ensemble statistical analysis.
Moreover, if the prior ensemble extends into the future, the result of the observational update can include an ensemble
statistical forecast of the state of the system based on past observations. This forecast only relies on the statistical dependence



between past and future as described by the prior ensemble, and as interpreted by the observational update algorithm. Such a 4D statistical analysis and forecast can be seen as an additional byproduct of the system, which could be obtained at negligible cost (as compared to the ensemble simulations).

Furthermore, the decoupling between model simulations and inverse methods substantially reduces the complexity of the numerical apparatus, which becomes more easily manageable and more flexible. Without the need to initialize the dynamical model, the ensemble analysis and forecast does not need to include the full state vector, but can concentrate on a few variables or diagnostics, possibly for a specific subregion, where the result is most needed. As illustrated in this paper, the possibility to reduce the dimension of the inverse problem opens new prospects in terms of inverse method, which can be more sophisticated,

and thus more able to deal accurately with complex prior ensemble statistics, nonlinear observation operators and non-Gaussian observation errors. For these reasons, there are certainly practical situations in which it would be interesting to append such a 4D statistical analysis and forecast to existing ensemble data assimilation systems. They may serve as a baseline to compare with the dynamical ensemble forecast, and as a possible substitute whenever useful.

The obvious shortcoming of this approach is that the complex nonlinear dynamical model is no more directly used to

constrain the solution, but only indirectly through the statistics of the prior ensemble. Moreover, if the prior ensemble has been run a long time from a realistic initial condition, the prior ensemble spread may be substantially larger than in classic data assimilation systems, with prior members possibly further away from the observations. In this respect, the approach is clearly suboptimal as compared to existing ensemble 4D analysis methods like the 4-Dimensional Ensemble Variational (4DEnVar, Buehner et al., 2013, see Lorenc, 2013, for a nomenclature of hybrid ensemble-variational methods) and the two-step Ensemble

smoother (see Van Leeuwen et al., 1996; Cosme et al., 2012, for a review of ensemble smoothers), which were developed in the framework of the well-known 4DVar and Kalman filter methods.

However, the approach proposed here also brings important advantages explained above (use of past and future observations, statistical forecast capability, simplification of the inverse problem,...), especially if the model is very sensitive to small imbalances in the initial condition and if it is difficult to produce an accurate ensemble analysis for all influential model state

variables (e.g. by lack of sufficient observations), as is often the case with ocean biogeochemical models. In such cases, the suboptimality of the approach (in the use of the dynamical constraint) can easily be counterbalanced by a better robustness and reliability.

The objective of this paper is to assess the strengths and weaknesses of this strategy by producing a 4D analysis and forecast of ecosystem variables and indicators for a small subregion ($10° \times 6.5°$) in the North Atlantic. The prior 40-member ensemble is

produced with a global $1/4°$ resolution NEMO/PISCES model configuration (prepared by Mercator Ocean International, later referred to as MOI), with stochastic parameterization of uncertainties. Observations are L3 product of chlorophyll data from ocean colour satellites. The results are illustrated for observed and non-observed variables, including ecosystem indicators, which are not model state variables. Probabilistic scores are used to evaluate the reliability and accuracy of the ensemble forecasts of chlorophyll concentration.

In addition, we show that the proposed strategy is compliant with a variety of observational update schemes similar to those in use today in operational systems. Two observational update algorithms are considered in the present paper to condition





the prior ensemble on the observations. The first one is the analysis step of the Ensemble Transform Kalman Filter (ETKF, Bishop et al., 2001) with domain localization (e.g., Janjic et al., 2011), using an implementation framework inherited from the SEEK filter (Pham et al., 1998; Testut et al., 2003). This is the same algorithm that is used in the MOI data asimilation

system (Lellouche et al., 2021), which illustrates the direct applicability of this approach in a real system. In this study, we just added time localization to our existing implementation to make it applicable to long time periods. The second one is the MCMC sampler recently developed by Brankart (2019). The idea here is to illustrate the potential benefit that can be obtained using a method that is more expensive, and still difficult to apply to the global multivariate system. It is worth pointing out once again that the aim of this article is not to compare the performance of the ETKF and MCMC schemes. Instead, we show

that the practical advantages of the proposed inversion method can benefit both schemes and we provide interpretations of their respective behaviours.

The paper is organized as follows. In section 2, we describe the ensemble simulation that has been performed to produce the prior ensemble. In section 3, we formulate the inverse problem that we are going to solve: region and variables of interest, observations, prior statistics. In section 4, we present the inverse methods (ETKF observational update algorithm and MCMC

sampler). In section 5, we illustrate the analysis and forecast results, including the forecast probabilistic scores.

## 2   Prior ensemble simulation

The purpose of this section is to present the stochastic NEMO/PISCES simulator, which defines the prior probability distribution for the evolution of the coupled physical/biogeochemical system. We first describe the deterministic NEMO/PISCES model (provided by MOI), from which we started, in section 2.1, then the stochastic parameterization transforming the deter-

ministic evolution of the system into a probability distribution in section 2.2, and finally the ensemble numerical simulation that has been performed to sample this probability distribution in section 2.3.

### 2.1   Model description

The physical component is based on the primitive equation model NEMO (Nucleus for European Modelling of the Ocean, Madec et al., 2017), with the configuration ORCA025 covering the world ocean at a $1/4°$ resolution and with 75 levels

along the vertical. The initial condition is specified using the GLORYS2V4 reanalysis data (doi:10.48670/moi-00024) for the beginning of 2013, and the atmospheric forcing is derived from the ERA5 reanalysis.

The biogeochemical component of the model is based on PISCES-v2 (Aumont et al., 2015), which includes 24 biogeochemical variables. There are 5 nutrients (nitrate, ammonium, phosphate, silicate and iron), two types of phytoplankton (nanophytoplankton and diatoms, with predictive variables for carbon, chlorophyll, iron and silicon), and two types of zooplankton

(microzooplankton and mesozooplankton, for which only the total biomass is modelled). Detritus are described by one variable for dissolved organic carbon and two variables for particulate organic matter (small and big particles, with a different sedimentation velocity). Among these variables, the initial condition for nutrients (phosphate, nitrate, silicate) and oxygen have been interpolated from the World Ocean Atlas 2018 (Garcia et al., 2018a, b); the initial condition for dissolved organic

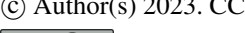



and inorganic matter, total alkalinity and dissolved iron are taken from the gridded data from the Global Ocean Data Analysis
Project (GLODAP v2, Olsen et al., 2020), while the other variables are initialized with constant values.

From the initial condition described above for the physical and biogeochemical components of the model, a 2-year determin-
istic spinup of the coupled model has been performed, starting on January 1, 2017, to obtain a stabilized and coherent initial
condition for the coupled model for our period of interest (year 2019).

## 2.2 Stochastic parameterization

The deterministic model configuration described above has been transformed into a probabilistic model by explicitly simulating
model uncertainties, using the stochastic parameterization approach proposed for NEMO in Brankart et al. (2015). More
precisely, three types of uncertainty have been introduced in the model:

1. *Uncertainties in the biogeochemical parameters.* Following the work of Garnier et al. (2016), 7 parameters have been
   perturbed by a time-dependent multiplicative noise with a lognormal probability distribution. These parameters have
been chosen not only because they are uncertain, but also because they have a large influence on the primary production
   in the model. They are: (1) the photosynthetic efficiency of nanophytoplankton; (2) the photosynthetic efficiency of
   diatoms; (3) the nanophytoplankton growth rate at $0°C$; (4) the sensitivity of phytoplankton growth rate to temperature;
   (5) the sensitivity of zooplankton grazing rate to temperature; (6) the dependence of nanophytoplankton growth on day
   length; (7) the dependence of diatoms growth on day length.

2. *Uncertainties due to unresolved scales in the biogeochemical tracers.* As a result of the nonlinear formulation of the
   biogeochemical model, unresolved fluctuations of the model variables can produce a substantial effect in the coarse-
   grained evolution equations, which is at least partly stochastic and which it is difficult to describe using a deterministic
   parameterization. Following Garnier et al. (2016), this effect has been parameterized by averaging the biogeochemical
   fluxes between model compartments over a set of fluctuations of the biogeochemical tracers. These fluctuations were
assumed proportional to the tracers themselves, with a time-dependent multiplicative noise.

3. *Location uncertainties.* Following the work of Leroux et al. (2022), uncertainties in the Lagrangian advection operator
   have been introduced in the model to simulate the effect of unresolved scales in the advection of physical and biogeo-
   chemical quantities. In practice, this is done by introducing a multiplicative noise in the metrics of the model grid, which
   correspond to simulating an additional uncertainty in the location of all model fields after each time step. This uncertainty
is thus expected to influence both the physical and biogeochemical components of the model.

These uncertainties are all simulated using two-dimensional maps of autoregressive processes (drawn independently for each
uncertainty and each parameter), whose characteristics are given in Table 1 (consistently with what was done in the papers
cited hereabove). Of course, we do not expect that they encompass all possible sources of uncertainty in the model. In the
paper, this assumption about uncertainties will be used as an attempt to make the stochastic simulator consistent with the





| Source of uncertainty | $\sigma$ | $\tau$ | $\rho$ |
|---|---|---|---|
| Biogeochemical parameters | 40% | 30 days | 5 grid points |
| Unresolved scales in tracers | 20% | 30 days | 5 grid points |
| Location uncertainties | 3% | 5 days | 5 grid points |

**Table 1.** Characteristics of the maps of autoregressive processes used to simulate each type of uncertainty: standard deviation ($\sigma$), correlation time scale ($\tau$) and correlation length scale ($\rho$).

available observations. However, the validity of the assumption cannot be checked for non-observed variables, as for instance the ecosystem indicators, as further explained in the discussion of the results in section 5.3.

### 2.3 Ensemble experiment

An ensemble experiment has been performed to sample the probability distribution described by the stochastic NEMO/PISCES simulator. The sample size has been set to 40 ensemble members, which amounts to performing 40 model simulations from 160 the same initial condition and with independent random processes in the stochastic parameterization.

The ensemble simulation has been performed for the whole year 2019 and the model results have been stored every day (at least for our region of interest) for both the physical and biogeochemical components.

### 3 Inverse problem

Using the statistics of the prior ensemble described in the previous section, the target is now to produce an ensemble analysis 165 and forecast of the evolution of the ecosystem. This is formulated as an inverse problem, in which the probability distribution described by the prior ensemble is conditioned on observations. The purpose of this section is to describe the problem that is going to be solved in the paper: (i) the subregion and variables of interest (in section 3.1), (ii) the Bayesian formulation of the problem (in section 3.2), (iii) the observations (in section 3.3), and (iv) the characteristics of the prior ensemble in the region of interest, with a comparison to observations (in section 3.4).

### 170 3.1 Region and variables of interest

In this study, the focus will primarily be on the analysis and forecast of the surface chlorophyll concentration, which is the observed quantity (see below). Second, we will examine how this result translates to other depth levels and to non-observed quantities, like zooplankton and particulate organic matter. Third, attempts will be made to see if more advanced diagnostics or indicators about the evolution of the ecosystem can be directly obtained as a solution of the 4D inverse problem: (i) the 175 timing of the bloom for phytoplankton and zooplankton (phenology), (ii) the part of the primary production that is converted into secondary production (vertically integrated trophic efficiency), and (iii) the part of the resulting organic material that is trapped in the ocean (downward flux of particulate organic matter at 100 m depth).





To illustrate the approach, the inverse problem will be limited to: (i) a small subregion in the North Atlantic, between $31°$W and $21°$W in longitude and between $44°$N and $50.5°$N in latitude, (ii) a 3-month time period between March 15 and June 15, 2019, including the spring bloom, (iii) a depth range between the ocean surface and 220 m depth, and (iv) a small subset of the PISCES state variables, together with a few diagnostic quantities (e.g. the vertically integrated trophic efficiency and the downward flux of particulate organic carbon, considered as 2 ecological indicators of interest, see section 5.3) that are introduced in the augmented state vector. In terms of size, this amounts to $40 \times 40$ grid points in the horizontal, 32 depth levels, 93 time steps (days), for typically 6 three-dimensional variables and 2 two-dimensional variables, so that the total size of the estimation vector $\mathbf{x}$ is $n = 40 \times 40 \times 93 \times (6 \times 32 + 2) = 28,867,200$. This size is kept small enough to make the 4D inverse problem easily tractable at reasonable numerical cost.

### 3.2 Formulation of the problem

The 4D inverse problem is formulated using the standard Bayesian approach. First, we assume that we have a prior probability distribution $p^b(\mathbf{x})$ for the estimation vector $\mathbf{x}$ (with dimension $n$ as defined above). In our case, this prior distribution is defined by the stochastic NEMO/PISCES simulator described in section 2. Second, we assume that we have a vector of observations $\mathbf{y}^o$, which is related to the true value $\mathbf{x}^t$ of the estimation vector by $\mathbf{y}^o = \mathcal{H}(\mathbf{x}^t) + \boldsymbol{\epsilon}^o$, where $\mathcal{H}$ is the observation operator and $\boldsymbol{\epsilon}^o$ is the observation error. The observation error is specified by the conditional probability distribution $p[\mathbf{y}^o|\mathcal{H}(\mathbf{x})]$, which describes the probability of an observation for a given value of $\mathbf{x}$. From these two inputs, we can obtain the posterior probability distribution $p^a(\mathbf{x})$ for the estimation vector (i.e. conditioned on observations) using the Bayes theorem:

$$p^a(\mathbf{x}) = p[\mathbf{x}|\mathbf{y}^o] \propto p^b(\mathbf{x}) \, p[\mathbf{y}^o|\mathcal{H}(\mathbf{x})] \tag{1}$$

Our purpose throughout this paper is to solve the 4D inverse problem by producing a sample of this posterior distribution.

For further reference in the paper, it is useful to take the logarithm of this equation and define the cost function $J(\mathbf{x})$:

$$J(\mathbf{x}) = J^b(\mathbf{x}) + J^o(\mathbf{x}) \tag{2}$$

where $J^b(\mathbf{x}) = -\log p^b(\mathbf{x})$ is the background cost function, $J^o(\mathbf{x}) = -\log p[\mathbf{y}^o|\mathcal{H}(\mathbf{x})]$ is the observation cost function, and $J(\mathbf{x}) = -\log p^a(\mathbf{x}) + K$, where $K$ is a non-important constant. Ratios in probability densities translate into differences in cost functions. The larger the cost function $J(\mathbf{x})$, the smaller the posterior probability.

The main difficulty to solve this problem comes from the large dimension of the estimation vector $\mathbf{x}$ ($n = 28,867,200$) and the small dimension of the prior ensemble ($m = 40$), so that specific methods with dedicated approximations are needed (see section 4). This will not go without a partial reformulation of the inverse problem. First, the undersampling of the prior distribution will always require solutions to avoid the spurious effect of non-significant long-range correlations, either by solving the problem locally (domain localization) or by adjusting the prior correlation structure (covariance localization). Second, assumptions will also be needed on the shape of the probability distributions. For instance, if we can assume that both $p^b(\mathbf{x})$ and $p[\mathbf{y}^o|\mathcal{H}(\mathbf{x})]$ are Gaussian and that $\mathcal{H}$ is linear, then both $J^b$ and $J^o$ are quadratic, so that $J$ is also quadratic, and a linear observational update algorithm can be used to obtain a sample of $p^a(\mathbf{x})$ from a sample of $p^b(\mathbf{x})$. Two options





that can potentially be activated in operational systems are considered here: one using a linear algorithm (ETKF), in which all distributions are assumed Gaussian, and one using a nonlinear iterative algorithm (MCMC sampler), in which only $p^b(\mathbf{x})$ need be assumed Gaussian (to generate appropriate random perturbations efficiently), but not $p[\mathbf{y}^o|\mathcal{H}(\mathbf{x})]$ and $p^a(\mathbf{x})$. Both will require applying a transformation operator (anamorphosis) to the original variables (which are not Gaussian).

### 3.3 Observations

The observations $\mathbf{y}^o$, which are used as conditions in the inverse problem (1), are surface chlorophyll concentrations derived from ocean colour satellites, as provided in the CMEMS catalogue (Globcolour L3 product). They are available as daily images at a $1/24°$ resolution, as illustrated in Fig. 1 between May 22 and May 27, 2019 for our region of interest. We see that the coverage is very partial in space and time as a result of the presence of clouds masking the ocean surface.

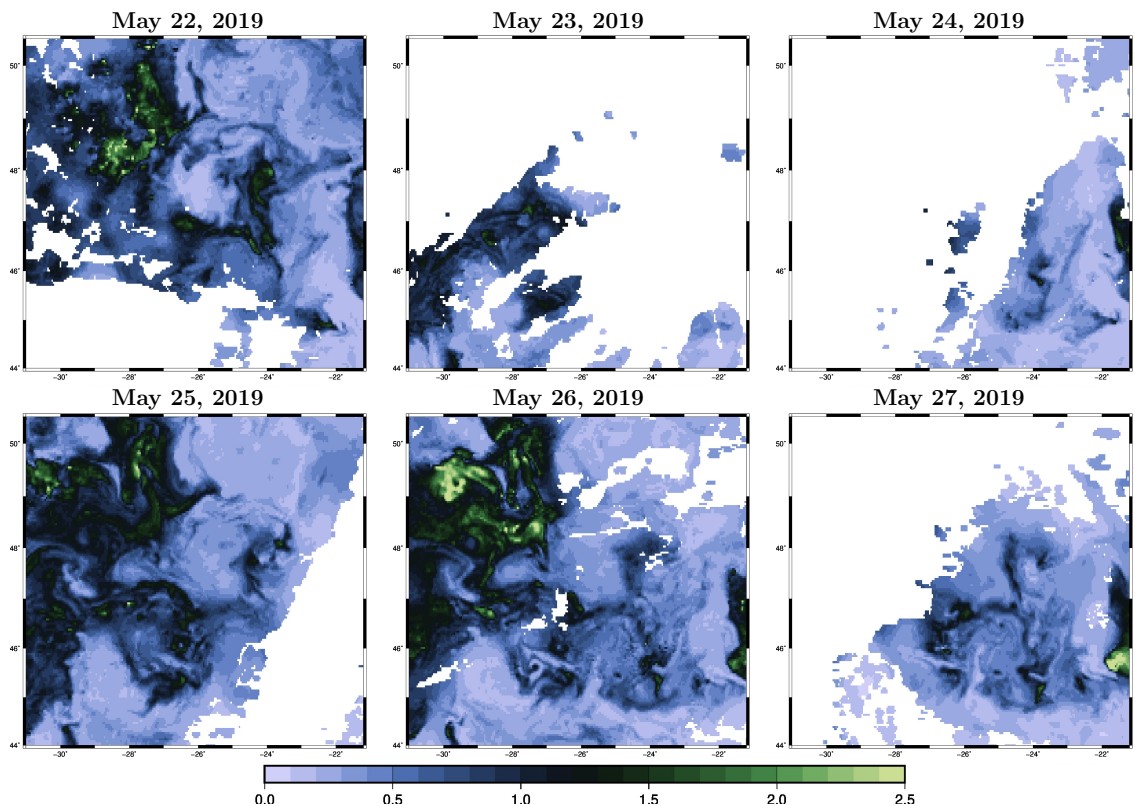

**Figure 1.** Observation of surface chlorophyll concentration (in mg/m$^3$, L3 ocean colour product) between May 22, 2009 and May 27, 2009.

Since chlorophyll concentration is one of the PISCES variable included in the estimation vector $\mathbf{x}$, the observation operator $\mathcal{H}$
is just a linear interpolation from the model grid to the location of observations. For the observation error probability distribution $p[\mathbf{y}^o|\mathcal{H}(\mathbf{x})]$, we assume lognormal marginal distributions, with a standard deviation equal to 30% of the true value of the



concentration, and we neglect observation error correlations. To make the assumption of zero observation error correlation more reasonable, we applied a data thinning to the original data, by subsampling the data by a factor 3 in each direction. With this reduction, the total size of the observation vector $\mathbf{y}^o$ over the 3 months of the experiment is $p = 182,837$. However, when used for verification purpose, we always keep the full resolution of the observations, and the total size of the observation vector is then $p' = 1,643,150$.

If the inverse method requires a Gaussian observation error probability distribution, then an approximation is required as explained in section 4 and in the appendices.

### 3.4 Prior ensemble in the region of interest

The prior distribution $p^b(\mathbf{x})$ in the inverse problem (1) is described by a sample provided by the ensemble simulation presented in section 2. Figure 2 illustrates the distribution obtained for the chlorophyll surface concentration in our region of interest for May 26, 2019. The figure displays 3 members of the ensemble (top panels) and 3 quantiles (20%, 50% and 80%) of the marginal ensemble distributions (bottom panels). For a 40-member ensemble, this means for instance that, at a given location, there are 8 members below the 20% quantile, 8 members above the 80% quantile, and 24 members in between.

From this figure, we can see that the spread of the prior ensemble, which results from the uncertainties embedded in the NEMO/PISCES simulator, is very substantial as compared to the median value (50% quantile). Ensemble members also display a variety of patterns, which are triggered by the space and time decorrelation of the stochastic perturbations. If we compare to the observations obtained for May 26 in Fig. 1 (middle bottom panel), we can see that the ensemble members have a comparable order of magnitude, and a large part of the observations fall in the range defined by the 20% and 80% quantiles.

Actually, to be consistent with the observations, the ensemble simulation should behave in such a way that 50% of the observations are above the median, 60% between the 20% and 80% quantiles, etc. More generally, an ensemble of size $m$ defines $m + 1$ intervals in which an observation can fall, and there should be an equal probability for the observation to fall in each of these intervals. This is the basis of the rank histogram approach to test the consistency of an ensemble simulation with observations. The histogram of all ranks of observations in the corresponding intervals defined by the ensemble should be flat (equal probability for an observation to fall in any of the intervals). More precisely, in presence of observation errors, the ensemble members must be randomly perturbed with a noise sampled from the observation error probability distribution before computing the ranks.

Figure 3 (left panel) shows the rank histogram obtained for the prior ensemble using all observations in our region of interest during the 3-month time period (i.e. with $p' = 1,643,150$ observations). We can see that this histogram is not flat: the ensemble simulation is here overdispersive. Uncertainties in the model have been overestimated: there are too many observations close to the median of the ensemble and not enough in the external intervals defined by the ensemble. It would certainly be better if the rank histogram could have been perfectly flat, but an underdispersive ensemble would have been much worse. Underestimating prior uncertainties would indeed mean that the dynamical ensemble simulation has not explored regions of the state space corresponding to the observed values, and the solution of the inverse problem would imply positionning the posterior probability distribution in these regions where the prior probability is zero and where we have no dynamical information about



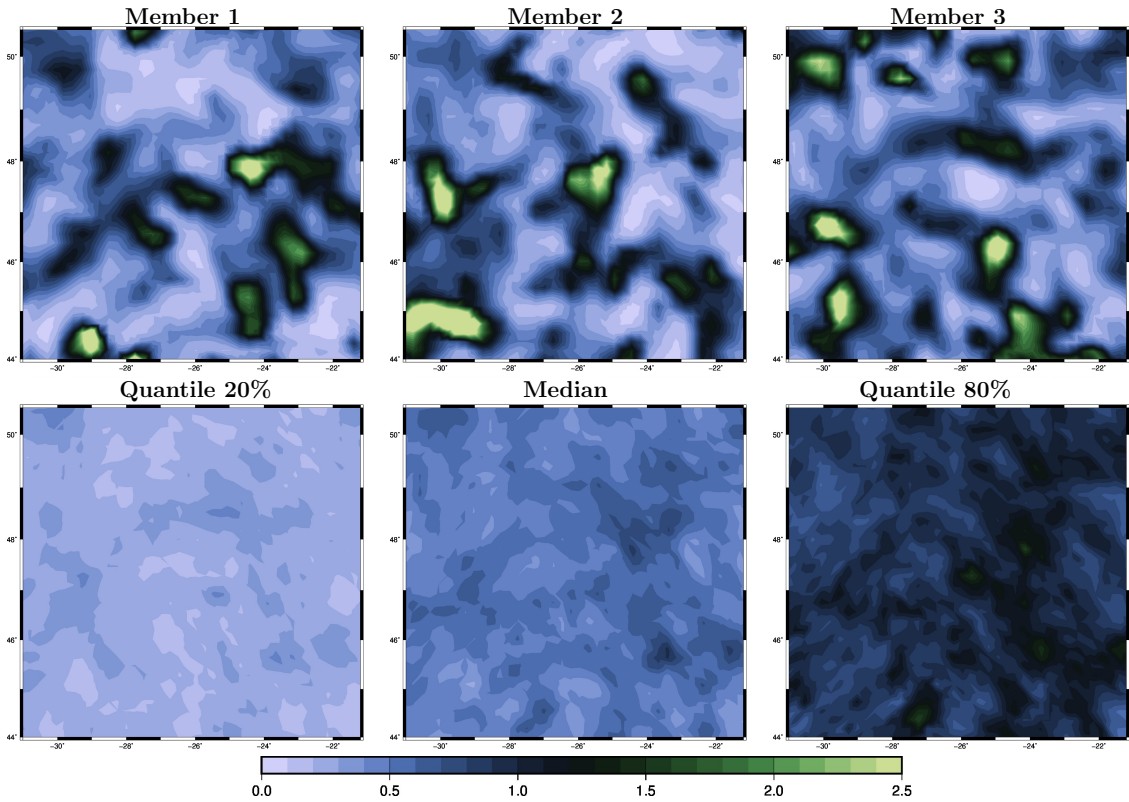

**Figure 2.** Prior ensemble. The figure displays 3 members of the ensemble (top panels) and 3 quantiles (20%, 50% and 80%) of the marginal ensemble distributions (bottom panels).

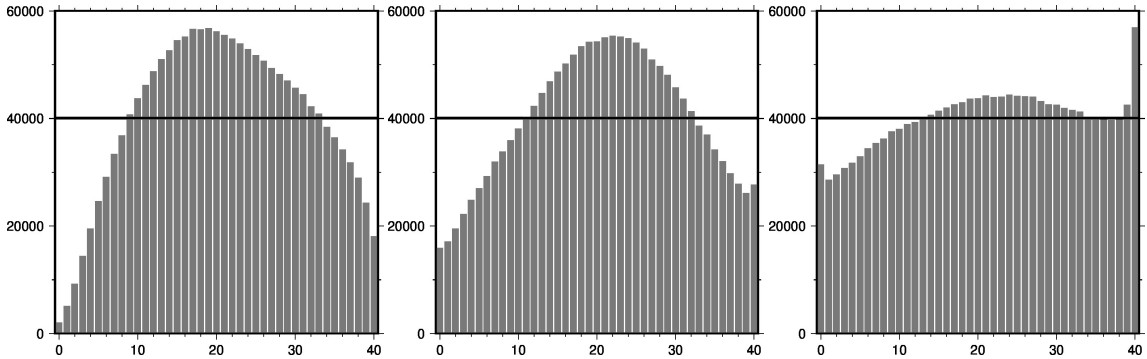

**Figure 3.** Rank histograms corresponding to the prior ensemble (left panel) and to the posterior ensemble, as obtained with the LETKF algorithm (middle panel) and the MCMC sampler (right panel). For the posterior ensembles, the histograms aggregate ranks from two simulations, one conditioned on observations from the odd days only, which is validated using observations from the even days, and another one conditioned on observations from the even days only, which is validated using observations from the odd days.



the behaviour of the system. Such extrapolation might be possible for the observed variable, but would be very hazardous for non-observed quantities.

It must be noted that, in the prior simulation described in section 2, there are other regions and/or seasons where the ensemble is still very underdispersive and/or biased, with most observations falling outside of the range of possibilities explored by the model. In the Atlantic, this occurs for instance in most of the subtropical gyre and in the Southern edge of the Gulf Stream extension. In such a situation, the methods used in this paper would not be effective, and it is first necessary to improve the description of uncertainties in the prior model simulation.

## 4  Inverse methods

To solve the inverse problem described in the previous section, we show that several methods can be used in a versatile way. In the present demonstration, two methods will be applied: (i) the first one is the analysis step of the Ensemble Transform Kalman Filter (ETKF) proposed in Bishop et al. (2001), with domain localization (e.g., Janjic et al., 2011), (ii) the second one is the implicitly localized MCMC sampler proposed in Brankart (2019). The two methods provide a solution to the same problem, and produce an updated ensemble, which is meant to be a sample of the posterior probability distribution (conditioned on the observations). But they use different algorithmic choices and different types of approximation, which are described in the rest of this section. Note that other updating methods could have been considered, such as EnKF, 3DVAR, etc. In section 4.1, we describe the general algorithms that are used to perform the observational update (a Kalman filter and an MCMC sampler) and the level of generality that they achieve. In sections 4.2 and 4.3, we highlight what the two schemes imply in terms of localization scheme and anamorphosis transformation, respectively.

### 4.1  Observational update algorithm

In Kalman filters (like the ETKF), the mean and covariance of the updated ensemble are computed from the mean and covariance of the prior ensemble using linear algebra formulas involving the observation vector, the observation operator and the observation error covariance matrix. This relies on the assumption that both prior and posterior distributions, as well as observation errors, are Gaussian, and that the observation operator is linear. If put in square-root form (as the ETKF), the scheme directly use and provide the square-root of the covariance matrix (possibly with low rank), thus ensemble members. Our specific implementation of this algorithm is inherited from the SEEK filter (Pham et al., 1998), which is another square-root filter, with an analysis step that can easily be made equivalent to that of the ETKF (especially once it has been adapted to allow for domain localization, as in Brankart et al., 2003; Testut et al., 2003).

By contrast, MCMC samplers are iterative methods, which converge towards a sample of the posterior probability distribution. Our particular implementation is a variant of the Metropolis/Hastings algorithm (see e.g. Robert and Casella, 2004), which is designed in such a way that the stochastic process, which generates the next element of the Markov chain, is in equilibrium with the probability distribution that must be sampled. This is obtained by an iteration in two steps: (i) one step to draw a random perturbation from a proposal probability distribution, and (ii) one step to accept or reject this new draw according




to the variation of the cost function. In this respect, our variant (Brankart, 2019) is somehow specific in the sense that (i) the proposal distribution is based on the prior ensemble (with covariance localization, see below), and (ii) only the observation

component of the cost function is needed to evaluate the acceptance criterion. Because of this, the method still requires that the prior probability distribution be Gaussian, but no assumption is made here on the shape of the posterior probability distribution, and on the observation constraint, which can be nonlocal, nonlinear, and even non-differentiable.

## 4.2 Localization

In ensemble Kalman filters, localization is used to alleviate the effect of a small ensemble size as compared to the number of

295 degrees of freedom in the system. In terms of covariance, the main negative effect of an insufficient sample size is that zero correlations are approximated by non-zero correlation, which can produce a substantial spurious effect on the results. Localization can be obtained with two different approaches: (i) domain localization (e.g., Janjic et al., 2011), in which the solution at a given location is computed using observations only within a specified neighbourhod (with a decrease of the observation influence with the distance to avoid discontinuities), and (ii) covariance localization (e.g., Houtekamer and Mitchell, 1998),

in which the ensemble covariance is transformed by a Schur product with a local-support correlation matrix (thus zeroing the long-range correlations). In both cases, it is assumed that the ensemble size is sufficient to describe the covariance structure when considered locally.

In square-root Kalman filters (like the ETKF and SEEK filters), covariance localization is difficult to apply because the ensemble covariance is never explicitly computed, only the square-root is available. A solution has nevertheless been proposed

by Bishop et al. (2017) in the framework of the ETKF, which consists in transforming each column of the covariance square-root by a Schur product with each column of a square-root of the localizing correlation matrix. The resulting matrix can be shown to be a square-root of the localized covariance matrix, which is what is needed by the square-root algorithm. However, with this approach, the number of columns of the square-root covariance is considerably increased (multiplied by the number of columns in the square-root of the localizing correlation matrix), and thus the cost of the resulting algorithm. This is why,

in the application below, we still use the more simple domain localization for the ETKF algorithm (as in the MOI operational system).

In the MCMC sampler, covariance localization can be introduced using a very similar approach as in the ETKF. If we assume that we can easily sample a zero-mean Gaussian distribution with the localizing correlation structure, then we can define the sampling of the proposal distribution as the Schur product of one member of the prior ensemble (anomaly with

315 respect to the mean) with one member of this localizing sample (with a normalized Gaussian random factor). From the same property used by Bishop et al. (2017) for the ETKF, the perturbation will have the same covariance as the prior ensemble, with covariance localization. But in this different context, the large increase in the number of directions of perturbation associated with localization becomes a benefit, since we want the perturbations to explore the estimation space as much as possible to fit the observations and produce the posterior sample. Moreover, if the localizing correlation can itself be expressed as the

320 Schur power of a specified correlation matrix, it is again possible to produce a very large sample (up to $10^8 - 10^{12}$) with the localizing correlation from a small sample (typically $10^2$) with the specified correlation, just by computing the Schur product





of a random combination of elements from the small sample (see Brankart, 2019, for more details). With the localization method, the sampling of the proposal distribution in many dimensions is thus reduced to the computation of a multiple Schur product with randomly selected members from a small sample, at a cost that is linear in the size of the system. This is important because this cost is usually quadratic, and thus often a limiting factor to the application of MCMC samplers in large-dimension problems.

### 4.3 Anamorphosis

Anamorphosis ($\mathcal{A}$) is a nonlinear transfomation that is applied to a model variable $x$ to transform its marginal probability distribution into a Gaussian distribution (with zero mean and unit variance). It is useful because many data assimilation method (like the two methods presented here) make the assumption of a Gaussian prior distribution. In this paper, we use the simple anamorphosis algorithm described in Brankart et al. (2012), which consists in remapping the quantiles of the marginal distribution of $x$ on the quantiles of the target Gaussian distribution, using a piecewise linear transformation (interpolating between the quantiles). The transformed variable $x' = \mathcal{A}(x)$ is then approximately Gaussian.

However, as explained above, Kalman filters also require that the observation operator is linear, and that the observation error is Gaussian. In this case, a similar transformation must also be applied to observations to keep the observation operator linear in the transformed variables. This can be done using the algorithm described in the appendices. This algorithm provides the right expected value and error variance for the transformed observation, but the detailed shape of the observation error probability distribution is lost by the transformation, since the observation error on this transformed observation must be assumed Gaussian. It is indeed impossible to find a transformation that ensures Gaussianity of both the prior and observational uncertainties, while keeping the observation operator linear.

By contrast, in the MCMC sampler, it is not required that the observation operator is linear, so that the observations do not need to be transformed. The original observation operator $\mathcal{H}$ needs only to be complemented by an inverse anamorphosis transformation $\mathcal{A}^{-1}$, to come back from the transformed vector $\mathbf{x}'$ to the orginal vector $\mathbf{x}$: $\mathcal{H}(\mathbf{x})$ is just replaced by $\mathcal{H}[\mathcal{A}^{-1}(\mathbf{x}')]$ This makes the use of anamorphosis much easier since only the estimation variables need to be transformed, not the observations. The observation constraint can be applied using the native observations, with the native observation operator and the native observation error probability distribution.

## 5 Results

In this section, we describe the solution obtained for the inverse problem formulated in section 3 using the methods presented in section 4. The focus will be on evaluating the reliability and accuracy of the ensemble analysis and forecast using dedicated probabilistic scores. The section is first devoted to the results obtained for the observed variable (the surface chlorophyll concentration), in terms of analysis (section 5.1) and forecast (section 5.2), and then extended to discuss the results obtained for non-observed variables and ecosystem indicators (section 5.3).



## 5.1 Analysis for observed variables

To illustrate first the idea of a single ensemble analysis extending over a long time period (3 months), Fig 4 shows time series of the surface chlorophyll concentration at the center of the region of interest (36°W, 44.25°N), as obtained with the two methods: the LETKF (localized ETKF) algorithm (top panel) and the MCMC sampler (bottom panel). The black curves represent 40 members of the prior ensemble (as produced by the NEMO/PISCES simulator) and the blue curves represent 40 members of the posterior ensemble (conditioned on all observations available in the period). We can see that the prior uncertainty in these time series is considerably reduced by the observations. For instance, at this location, the timing of the spring bloom was left very uncertain by the NEMO/PISCES simulation, probably because it is very dependent to the uncertain parameters that have been perturbed. But, even if partial and imperfectly accurate, the observations contain a lot of information about this, so that the variety of possibilities left in the posterior ensemble is much smaller (with both methods). On this respect, it must already be noted that the solution at a given time is influenced by both past and future observations. This is an important advantage over sequential ensemble filters, in which only past observations can influence the solution obtained at a given time.

Second, to illustrate the resulting maps of surface chlorophyll concentration, we choose to focus on May 26, 2019, when there is a good coverage of observations (see Fig. 1), with which the results can be compared. For this date, Fig. 5 (LETKF algorithm) and Fig. 6 (MCMC sampler) display 3 members of the posterior ensemble (top panels) and 3 quantiles (20%, 50% and 80%) of the posterior marginal ensemble distributions (bottom panels). This gives an idea of individual possibilities and of the spread of the ensemble, and can be directly compared to the same information provided for the prior ensemble in Fig. 2. Again, we can see that, with both methods, the spread of the ensemble is considerably reduced by the observation constraint, with all posterior members getting closer to the observed situation. Even if there are differences between methods (see discussion below), most observed structures are present in the two ensemble analyses. In both cases, however, there remains a variety of possibilities in terms of location and amplitude of the structures, as a result of the substantial uncertainty in the observations (a 30% observation error standard deviation). At first glance, despite the use of completely different algorithms (LETKF and MCMC sampler), the two ensemble analyses are generally reasonable and quite similar. They can thus be discussed together in more details.

### 5.1.1 Space and time localization.

Regarding the formulation of the inverse problem, the main difference between the two solutions is the localization scheme, which also contains the only free parameters of the inverse algorithm (i.e. not included in the definition of the problem in section 3). The localization parameters used in our experiments are given in Table 2. In both cases, we have to specify a length scale and a time scale, but there is no direct correpondence between the parameters used in the two algorithms because their behaviour is not the same. In principle, they must be set so that the effect of remote observations vanishes when the ensemble correlation becomes non-significant, but this always requires some additional tuning.

Not enough localization means that remote observations keep too much influence: the fit to local observation is lost, the ensemble spread is too small, and the solution thus becomes unreliable. Too much localization means that relevant remote





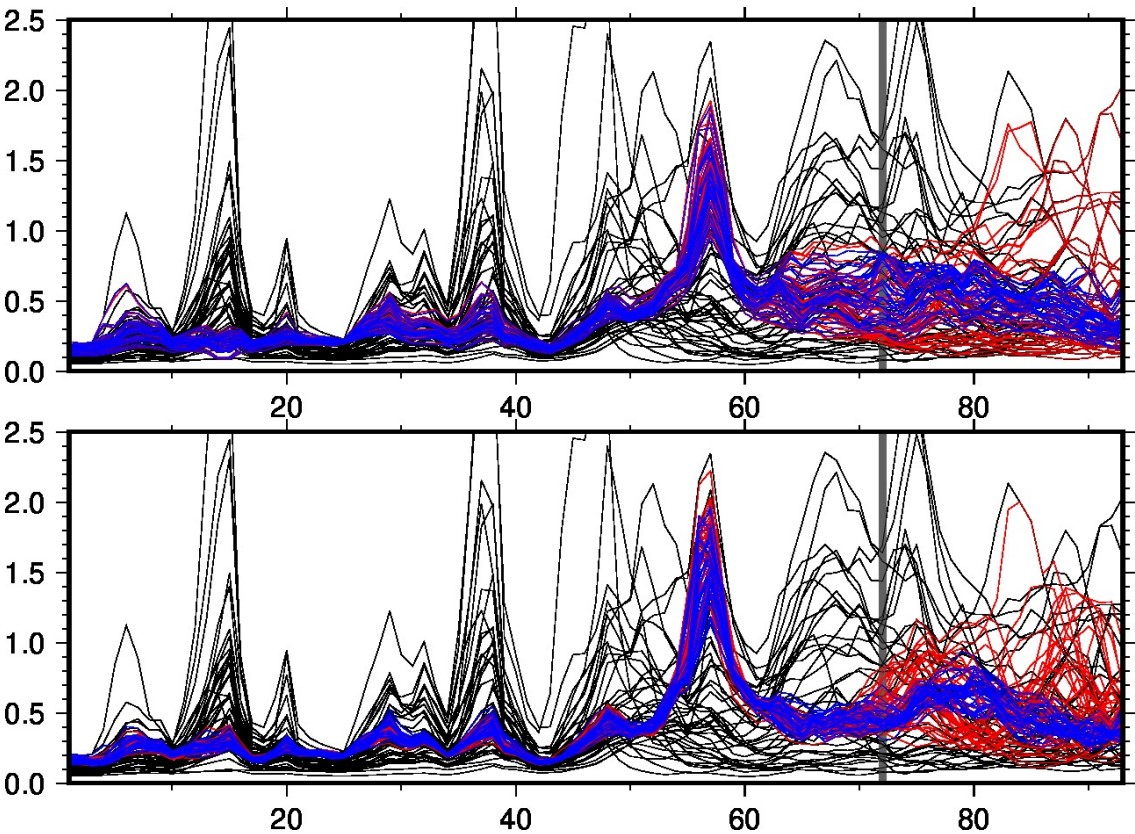

**Figure 4.** Ensemble surface chlorophyll time series at $36°$W, $44.25°$N. The black curves represent 40 members of the prior ensemble (as produced by the NEMO/PISCES simulator), the blue curves represent 40 members of the ensemble analysis (using observations for the entire period), and the red curves represent 40 members of the ensemble forecast (using only observations until May 25). The forecast thus starts on May 25 (vertical grey line). These results are shown for the LETKF algorithm (top panel) and the MCMC sampler (bottom panel).

observation is missed, the ensemble spread is too large and the solution becomes inaccurate. In our experiments, localization has been tuned heuristically to obtain a good compromise on the scores presented below: a sufficient ensemble spread to keep the solution reliable (rank histogram), a good fit to independent observations (probabilistic score) in the analysis and in the forecast. Regarding the local structure of the solution, we can see, by comparing Figs. 5 and 6, that covariance localization

tends to be more respectful of the local correlation structure of the prior ensemble (by construction), while domain localization can trigger small scales that are not present in the prior ensemble. There are situations in which this can make a big difference, but this may not be so important in the present application.





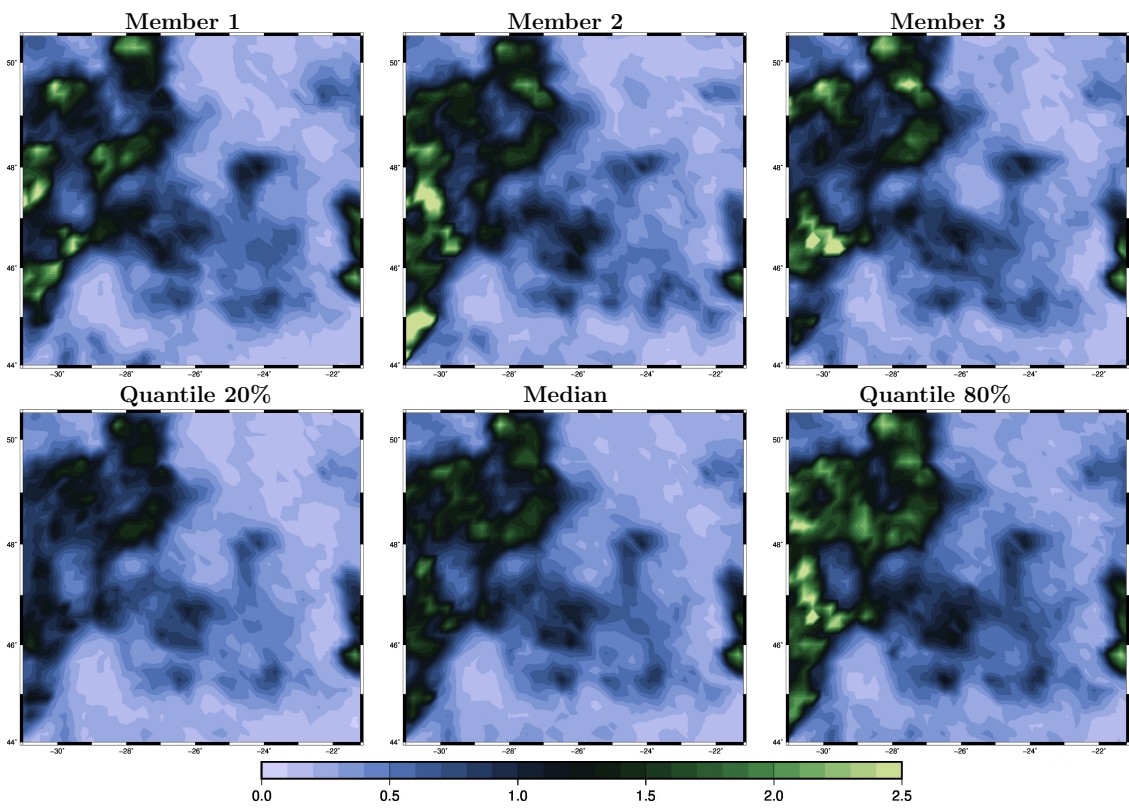

**Figure 5.** Ensemble analysis obtained with the LETKF algorithm. The figure displays 3 members of the ensemble (top panels) and 3 quantiles (20%, 50% and 80%) of the marginal ensemble distributions (bottom panels).

| Method | Parameter | Space | Time |
|---|---|---|---|
| LETKF | influence radius | 4.5 grid points | 7.5 days |
| algorithm | cut-off radius | 12 grid points | 20 days |
| MCMC | localizing sample decorrelation scale | 0.8 degrees | 7.5 days |
| sampler | localization decorrelation scale | 0.33 degrees | 3.1 days |

**Table 2.** Localization parameters used in the LETKF algorithm and in the MCMC sampler. In the LETKF algorithm, the localizing function is set to be isotropic (with a Gaussian-like shape) in the metrics defined by the grid of the model. In the MCMC sampler, the localizing correlation is set to be isotropic on the sphere (degrees are along great circles); the localizing sample is generated using a Gaussian spectrum in the basis of the spherical harmonics (in the horizontal) and a Gaussian spectrum in the basis of the harmonic functions (in time), so that the sample correlation structure is also Gaussian, and the localizing correlation structure as well. There is no localization along the vertical coordinate.



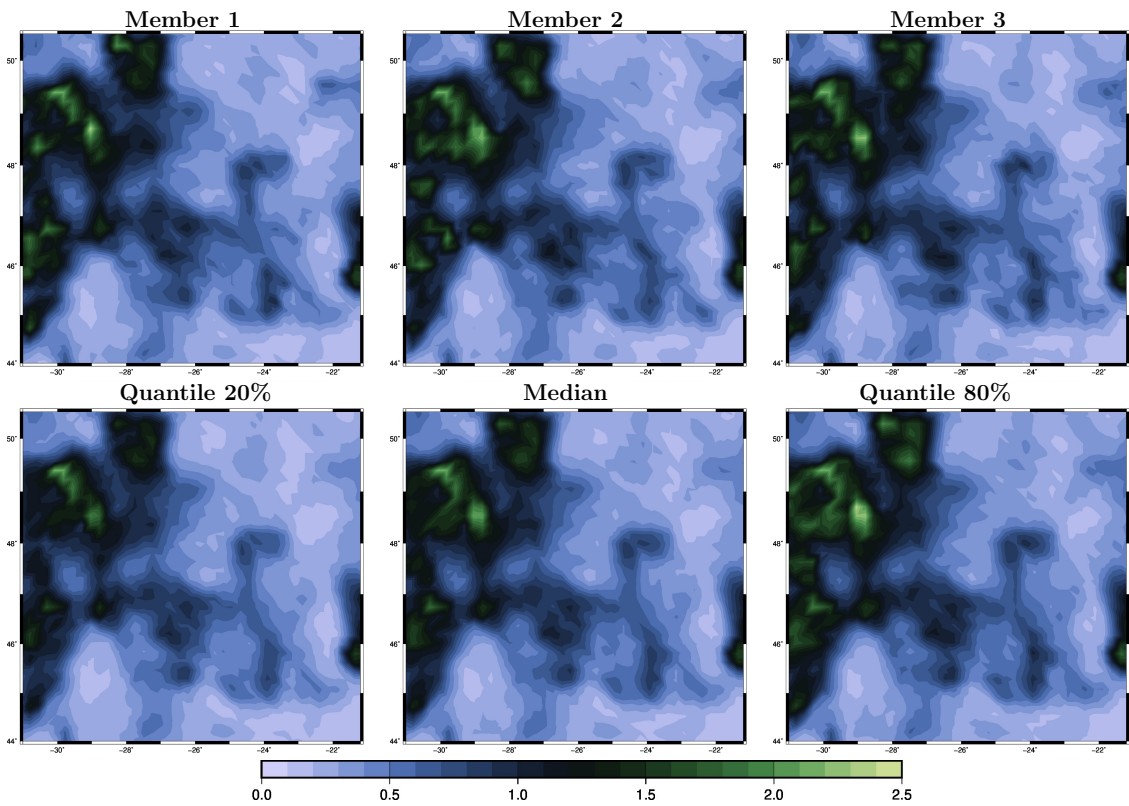

**Figure 6.** Ensemble analysis obtained with the MCMC sampler. The figure displays 3 members of the ensemble (top panels) and 3 quantiles (20%, 50% and 80%) of the marginal ensemble distributions (bottom panels).

### 5.1.2 Reliability.

The most important property of the analysis results is the reliability of the ensemble, i.e. the consistency with independent
verification data. This can be evaluated using rank histograms as explained in section 3.4 for the prior ensemble. To keep
independent surface chlorophyll observations, we performed two additional analysis experiments: one using observations from
the odd days only, which we can validate using observations from the even days, and another one using observations from the
even days only, which we can validate using observations from the odd days. By aggregating the ranks of the verification data
from these two experiments, we can obtain a rank histogram based on all available observations, as was done for the prior
ensemble.

Figure 3 displays the rank histogram obtained for the ensemble analysis performed with the LETKF algorithm (middle
panel) and with the MCMC sampler (right panel). In both cases, the resulting ensemble is not underdispersive, except for a
very small excess of observations above the maximum of the ensemble in the case of the MCMC sampler. Underestimating the
posterior uncertainty is never a good idea, and this can be directly avoided here by tuning the localization parameters until the





| Experiment | Reliability | Resolution | Total CRPS |
|---|---|---|---|
| Prior ensemble | $4.9 \times 10^{-3}$ | 0.1070 | 0.1119 |
| LETKF ensemble analysis | $1.0 \times 10^{-3}$ | 0.0727 | 0.0737 |
| MCMC sampler ensemble analysis | $0.2 \times 10^{-3}$ | 0.0712 | 0.0714 |

**Table 3.** Global CRPS score (in mg/m$^3$), using all available surface chlorophyll observations over the full time period of the experiments. The scores of the analyses aggregate the scores from two simulations, one using observations from the odd days only, which is validated using observations from the even days, and another one using observations from the even days only, which is validated using observations from the odd days.

posterior ensemble spread is sufficient (which is always possible, as long as the prior ensemble is itself reliable). Regarding this particular diagnostic, the localization of the LETKF could have been tuned towards less localization to decrease the spread of the posterior ensemble and make the overall rank histogram flatter. However, it must be kept in mind that this diagnostic aggregate a variety of different situations in space and time, and that other diagnostics also matter (e.g. the reliability of the forecast, see below).

The reliability of the results can also be assessed using the CRPS score (reliability component, as shown in Table 3, see definition below). This is done again by aggregating the scores from two simulations, using observations from odd days and even days, respectively. The reliability score obtained with the MCMC sampler ($0.2 \times 10^{-3}$ mg/m$^3$) is here better than that obtained with the LETKF algorithm ($1.0 \times 10^{-3}$ mg/m$^3$), but the two results can be considered as sufficiently good, as a direct result of the tuning of the localization parameters. On this respect, it must also be noted that more accurate observations would

make a more stringent test of reliability. With large observation uncertainties, the necessary conditions of reliability tested with these scores are more easily achieved.

### 5.1.3 Probabilistic scores.

To further compare the ensemble analysis with verification data, we must not only check the consistency between the results and the data, but also quantify the amount of information that the analysis provides about the data. This is done here using the

CRPS score, which is defined from the misfit between the marginal cumulative probability distribution of the ensemble (cdf, considered as a stepwise function increasing by $1/m$ at the value of each ensemble member) and the cdf associated with the corresponding observation (considered as a Heaviside function, increasing by 1 at the value of the observation). The CRPS score can be decomposed as the sum of a reliability component (characterizing the consistency between the ensemble and the observations) and a resolution component (characterizing the amount of information provided by the ensemble). Roughly

speaking, if the misfit is due to observations falling systematically outside of the range of the ensemble, it will contribute to the reliability component of the score. Conversely, if the misfit is due to the spread of the ensemble, it will contribute to the resolution component of the score.





| Size of the estimation vector | $28,867,200$ |
|---|---|
| Size of the observation vector | $182,837$ |
| Size of the prior ensemble | 40 |
| Size of the updated ensemble | 40 |
| Number of processors used | 640 |
| Clocktime for the LETKF analysis | 271 s |
| Clocktime for the MCMC sampler analysis | 1761 s |

**Table 4.** Dimension of the problem and cost of the analysis experiments. The time to read input data and write output data is not included in the clocktime.

Table 3 provides the value of the score obtained for the prior ensemble and for the analysis obtained by each of the two methods. Again, in the case of the analyses, the score is an aggregate computed from two different experiments, so that

only independent observations are used to evaluate the scores. From the table, we see that the total score and the resolution component of the score are very similar with the two methods. Moreover, it must be noted that the contributions to this score are quite heterogeneous in space and time, with none of the methods systematically outperforming the other in terms of CRPS misfit.

### 5.1.4 Numerical cost.

Table 4 provides the numerical cost of the two analysis experiments (with the LETKF and the MCMC sampler). A comparison of the computational complexity of the two algorithms as a function of the size of the problem is presented in Brankart (2019). It is here just briefly particularized to our application. From this table, we can see that, as applied in this study, the MCMC sampler is about 6.5 times more expensive than the LETKF algorithm. But this directly depends on choices that have been done in terms of parameterization and implementation.

In both algorithms, the computational complexity is linear in the size of the estimation vector ($n$) and in the size of the observation vector ($p$), so that this has no influence of their relative cost. In the LETKF, the cost is quadratic in the size of the updated ensemble, which must be the same as the size of the prior ensemble ($m$). Moreover, the cost is also proportional to the volume of the space/time domains used by the localization algorithm (defined by the cut-off length scale and time scale in Table 2). For given influence scales, this domain is difficult to reduce without risking space/time discontinuities in the resulting

fields. In the MCMC sampler, the cost is linear in the size of the updated ensemble ($m'$), which can be different from the size of the prior ensemble ($m$). This is an advantage because it is possible to compute only a few members at a much smaller cost (for example 8 members at a cost 5 times smaller) and then compute more and more members as deemed useful for a given application (for example 200 members at a cost 5 times larger). But the cost is also proportional to the number of iterations $N$ needed to reach the solution. This makes the cost of the MCMC sampler much less predictable than the cost of the

LETKF algorithm, because this depends on the number of degrees of freedom that must be controlled and the level of accuracy




required. In the experiments performed for this paper, $N$ is set to $30,000$ to be sure that the convergence has been reached. But experiments done with $N = 10,000$ iterations do not show a very substantial difference in the scores and in the results; the distance to the observations is only slightly larger. And the cost is reduced to a third of what is given in Table 4).

In terms of implementation, it must also be noted that the MCMC sampler is much easier to code and parallelize that the LETKF algorithm. Providing that the ensemble equivalent to the observations is provided as an input, each processor can deal separately with a block of the state vector and a block of the observation vector, with almost no communication between them, except the summation of the distributed components of the cost function (and a few other scalar quantities like the random coefficient to the global perturbation). This is possible in this algorithm because the inverse problem is solved globally and because covariance localization is obtained implicitly through global Schur products, computed separately in estimation space (to iterate the solution, with random perturbations) and in observation space (to compute the corresponding perturbations and evaluate the cost function).

## 5.2 Statistical forecast

If now we reduce the observation vector to include only observations before a given time $t_0$, the result of the inverse problem can be interpreted as an ensemble analysis before $t_0$ and an ensemble forecast after $t_0$. This forecast is a time extrapolation based on the statistics of the prior ensemble, with space and time localization. In our case, it is a nonlinear statistical forecast, since it is based on the linear space/time correlation structure for nonlinearly transformed variables (by anamorphosis). Figure 4 (red curves) shows the result obtained with the LETKF algorithm (top panel) and the MCMC sampler (bottom panel), when the last date of observation is May 25, 2019 (as represented by the vertical grey line in the figure). In both cases, the spread of the posterior ensemble increases during the forecast, and starts increasing before $t_0$ when future observations starts lacking. At $t_0$ indeed, only half of the observations used in the analysis are available (as would also be the case for the ensemble analysis in a sequential Kalman filter).

Regarding the future ($t > t_0$), we see that extrapolation in time is much more sensitive to the particulars of the localization method. In the LETKF algorithm (with domain localization), the problem is solved locally and separately for each time (and each horizontal location) and the influence of the past observations (before $t_0$) decreases with time as a result of the time decorrelation present in the prior ensemble and the superimposed localizing influence functions (as parameterized by the influence radii). This influence is then cut off when the local domain does not include $t_0$ anymore (in our case, at $t_0 + 20$ days, when the posterior members become exactly equal to the prior members). In the MCMC sampler (with covariance localization), the problem is solved globally for the whole time period and the influence of past observations (before $t_0$) decreases with time as a result of the time decorrelation present in an augmented version of the prior ensemble, whose covariance is localized by a Schur product with a localizing correlation function. This influence thus vanishes with time, but there is no cut-off time here, since there is no bound to the domain of influence. These differences explain why the forecast can behave somewhat differently in the solution produced by the two methods.





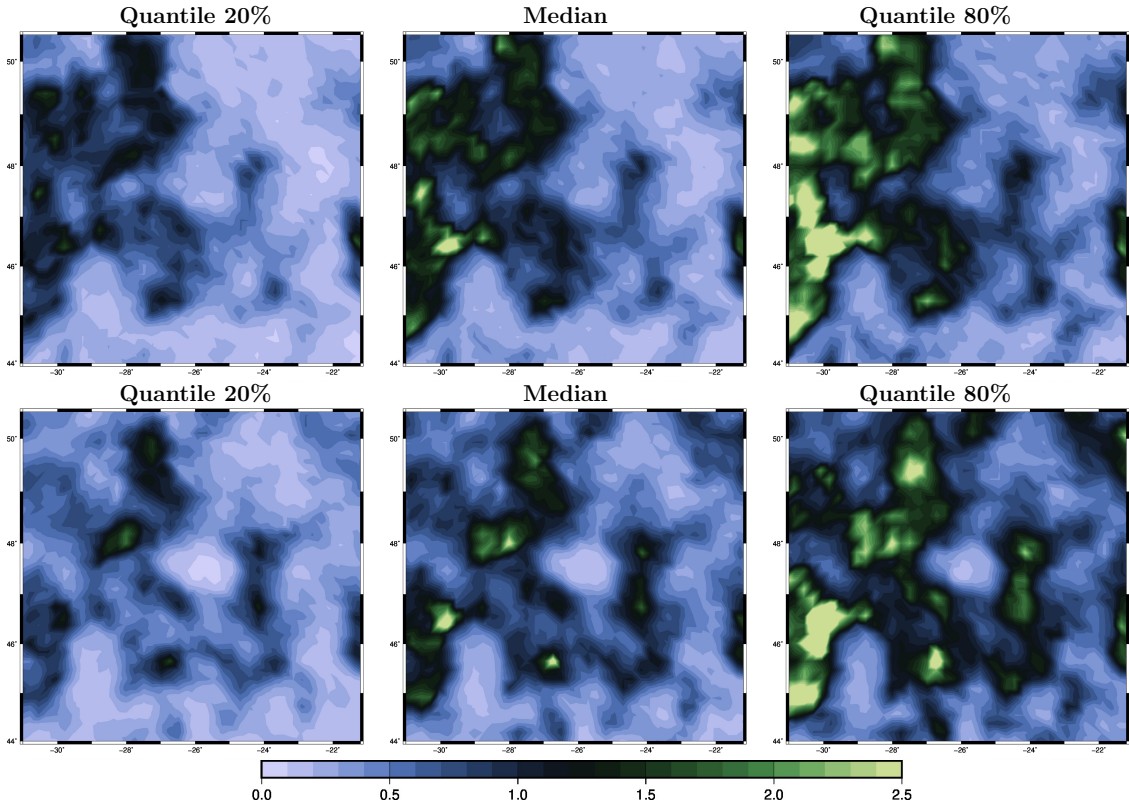

**Figure 7.** Ensemble forecast obtained with the LETKF algorithm. The figure displays 3 quantiles (20%, 50% and 80%) of the marginal ensemble distributions, as obtained for the 1-day forecast (top panels) and the 4-day forecast (bottom panels).

### 5.2.1 Forecast accuracy.

To further evaluate the forecast, we consider two ensemble forecasts obtained for the same date (May 26, 2019), but with a
different time lag with respect to the last observation. The first one is a 1-day forecast, with the last observation on May 25, 2019 (bottom left panel in Fig. 1), and the second one is a 4-day forecast, with the last observation on May 22, 2019 (top left panel in Fig. 1). The date of the forecast (May 26) is chosen so that the results can be directly compared to the observations in Fig. 1 (bottom middle panel), to the prior ensemble in Fig. 2, and to the ensemble analyses in Fig. 5 (LETKF algorithm) and in Fig. 6 (MCMC sampler). The ensemble forecast is described by 3 quantiles (20%, 50% and 80%) of the marginal distributions,
as shown in Fig. 7 (for the LETKF algorithm) and in Fig. 8 (for the MCMC sampler). The two figures include the 1-day forecast (top panels) and the 4-day forecast (bottom panels).

In these figures, we can see that the accuracy of the forecast is deteriorating with time, and the spread of the ensemble is increasing accordingly. After one day, the ensemble forecast is still very similar to the ensemble analysis and the observations (not used anymore in the inversion problem). Only a few structures are missing from the median and the ensemble spread is



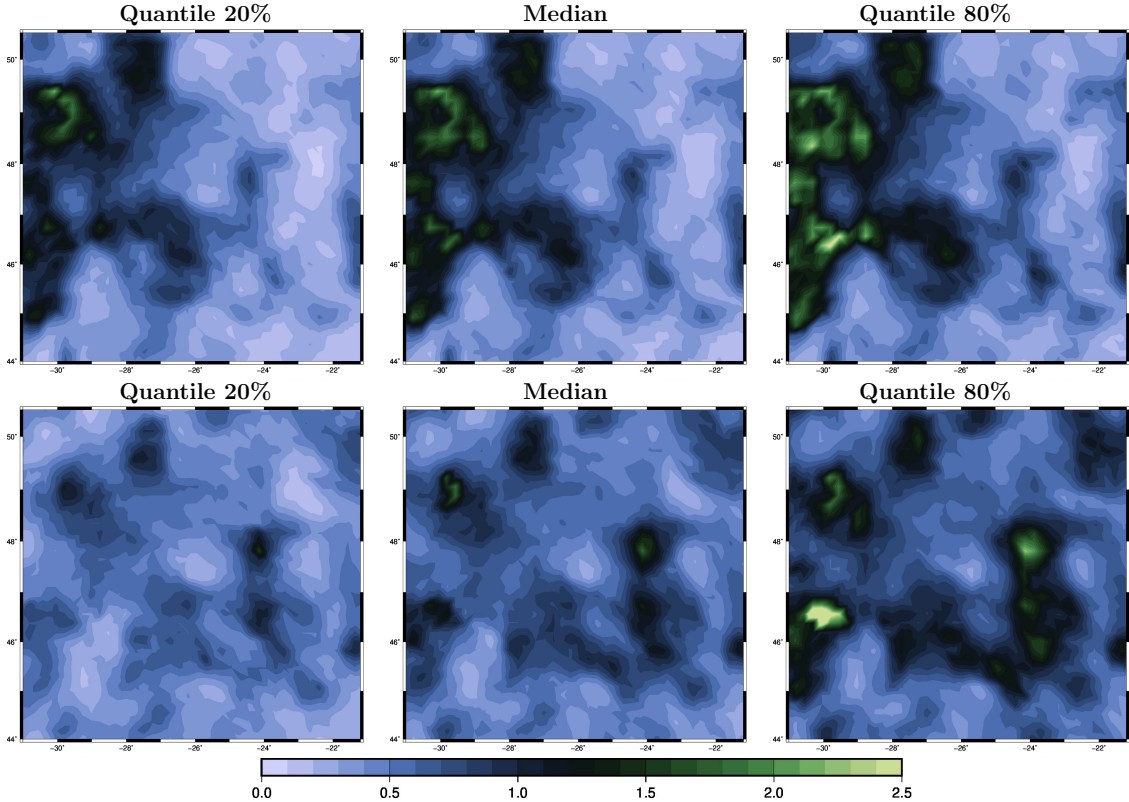

**Figure 8.** Ensemble forecast obtained with the MCMC sampler. The figure displays 3 quantiles (20%, 50% and 80%) of the marginal ensemble distributions, as obtained for the 1-day forecast (top panels) and the 4-day forecast (bottom panels).

not so much larger (see the scores below for a more precise quantification). After four days, the situation becomes more fuzzy. The forecast still contains valuable information, we can still recognize the main patterns of the observed surface chlorophyll concentration, but most local structures are missed or not correctly positioned. The posterior uncertainty described by the ensemble spread is also substantially larger, but still much smaller than the prior uncertainty. These general features of the 1-day and 4-day ensemble forecasts are shared by the two methods, but there are also differences in their behaviour. On the one

hand, the LETKF algorithm tends to keep more small scale structures and extreme chlorophyll concentrations (both low and high values), which are inherently more uncertain and require a larger spread. On the other hand, the MCMC sampler tends to produce a smoother solution (consistently with the prior ensemble) with less extreme values, which leads to sometimes missing the possibility of small scale structures in the forecast. This different behaviour of the two methods can be attributed to the difference in the space/time localization algorithm.

About the respective accuracy of the 1-day and 4-day forecasts, it is important to remark that the conclusions drawn above are not general. They depend on the correlation structure of the prior ensemble, which may be very heterogenous in space and





time, and on the availability of observations. For instance, in our case study, if we look at the observation coverage in Fig. 1, we see that we have one good observation coverage 1 day before May 26 and another one 4 days before May 26, which is why we decided to illustrate the results with a 1-day and a 4-day forecast. In this example, the 2-day forecast or the 3-day forecast would have been no better than the 4-day forecast since there is only very few observations available on May 23 and May 24. The capacity to make a forecast is obviously dependent on the availability of observations. In addition, it must be noted that our case study is not strictly speaking a forecast, since we used a prior ensemble simulation in which the atmospheric forcing is based on a reanalysis. For our experiment to be a real forecast, the prior simulation should have used an atmospheric forecast rather than a reanalysis. This would not be a difficulty in an operational context, and it is not likely to make a big difference in the illustration presented in this paper (at least for the short-term forecast).

### 5.2.2 Probabilistic scores.

Table 5 provides a more quantitative assessment of the forecast, with the probabilistic scores obtained for May 26, 2019, i.e. using all surface chlorophyll observations available on that day. The scores are given for the 1-day and 4-day forecasts, and compared to the scores obtained for the prior ensemble and the analysis, always on the same day. In addition to the CRPS score (reliability and resolution), we have also computed the RCRV score (reduced centered random variable), which is a measure of the reliability of the ensemble (and thus says nothing about the resolution) in terms of bias and spread (Candille et al., 2007; Candille et al., 2015). More precisely, the reduced variable is defined as the misfit between each piece of verification data and the corresponding ensemble mean, and then normalized by the ensemble standard deviation. If the ensemble is reliable, the expected value of this reduced variable (bias) must be 0 and its standard deviation (spread) must be 1 or 100%. If the bias is positive (resp. negative), this means that the ensemble is systematically too small (resp. too large) as compared to the verification data. If the spread of the reduced variable is too large, above 100% (resp. below 100%), this means that the spread of the ensemble is too small (resp. too large) to be consistent with the verification data. As for the CRPS, if the verification data are observations, the ensemble must be perturbed by the observation error before computing the score.

The figures of the scores provided in Table 5 show first that the prior ensemble and the analyses (LETKF and MCMC) are quite reliable for that date, with biases well below 10% of the ensemble spread, and a quite accurate ensemble spread, as confirmed by the low value of the CRPS reliability component. The largest misfit to reliability is in the LETKF analysis, which is somewhat overdispersive. Regarding the forecast, the 1-day forecast is still as reliable as the analysis, except for the MCMC 1-day forecast, which is somewhat underdispersive (134% RCRV spread). In terms of resolution, with a score of about 0.13 mg/m$^3$, the 1-day forecast is not very far from the analysis (about 0.1 mg/m$^3$) and still much better than the prior ensemble (about 0.25 mg/m$^3$). By contrast, in the 4-day forecast, both reliability and resolution degrade substantially. The bias grows to about $-16\%$ (for the LETKF algorithm) and $-8\%$ (for the MCMC sampler), and the MCMC ensemble forecast becomes even more underdispersive (165% RCRV spread). This is confirmed by their larger CRPS reliability score, which are similar but presumably for different reasons (one because of the larger bias, and the other because of the underdispersion of the ensemble). As already observed in Figs. 7 and 8, the forecast also contains much less information than the analysis, with a CRPS resolution





| Method | Experiment | CRPS | | RCRV | |
|---|---|---|---|---|---|
| | | Reliability | Resolution | Bias | Spread |
| Prior ensemble | | $1.0 \times 10^{-3}$ | 0.251 | 4.8 | 107.6 |
| LETKF algorithm | analysis | $5.8 \times 10^{-3}$ | 0.098 | 6.3 | 78.8 |
| | 1-day forecast | $4.9 \times 10^{-3}$ | 0.129 | -0.8 | 100.3 |
| | 4-day forecast | $9.8 \times 10^{-3}$ | 0.184 | -16.1 | 113.2 |
| | biased analysis | $13.4 \times 10^{-3}$ | 0.100 | 34.3 | 88.2 |
| MCMC sampler | analysis | $1.6 \times 10^{-3}$ | 0.099 | 7.6 | 102.6 |
| | 1-day forecast | $1.0 \times 10^{-3}$ | 0.131 | 3.8 | 134.1 |
| | 4-day forecast | $9.5 \times 10^{-3}$ | 0.204 | -8.3 | 165.1 |

**Table 5.** CRPS score (in mg/m$^3$) and RCRV score (in %) for May 26, 2019. Since this is is an odd day in the time sequence, the scores of the analyses come from the experiment using observations from the even days only, so that the observations used for validation are independent.

growing to about 0.18 mg/m$^3$ for the LETKF algorithm and 0.20 mg/m$^3$ for the MCMC sampler, but still substantially better than the prior ensemble (about 0.25 mg/m$^3$).

In order to illustrate the importance of correctly applying the observation constraint when observation errors are non-Gaussian, we have added one line of scores in Table 5 for the LETKF algorithm (named "biased analysis"). In this "biased analysis", the algorithm is exactly the same except that the anamorphosis transformation of the observations is performed using the simplifed algorithm in Annex A1, rather than the more general algorithm in Annex A2, which is much needed in our application. With the simplified algorithm, there is indeed a strong bias on the ensemble analysis (about one third of the ensemble spread), in which the chlorophyll concentration is systematically too small as compared to the observations. The CRPS reliability score is also substantially increased, but the nature of the problem is less apparent with this score. In our problem, this spurious effect is large because the observation errors are large, but it is important to remain cautious when a Gaussian approximation of non-Gaussian observation errors is needed, as in Kalman filters (with or without anamorphosis). There is no such difficulty with the MCMC sampler, in which the observation constraint can be applied using the native observation error probability distribution. This special care about observation errors was only needed on the LETKF side to make the level of reliability of the two solutions coincide.

## 5.3 Ecosystem indicators

Up to now, the ensemble analysis and forecast have been diagnosed in terms of the observed variable only. In the following, we move to the diagnostic of non-observed quantities, starting with variables that can be expected to be well controlled by the observations, towards ecosystem indicators that are more likely to depend on uncontrolled processes: (i) the phenology of the bloom, (ii) the vertically integrated trophic efficiency, and (iii) the downward flux of particulate organic matter at 100 m depth.



### 5.3.1  Phenology.

As a simple characterization of the phenology of the spring bloom (at a given location in 3D), we use the first date at which the chlorophyll concentration reaches half of its maximum value over the whole time period. This definition has the advantage of being as closely related to the observations as possible, and of not being too sensitive to small uncertainties in the value of the concentrations. Unlike the maximum, it is indeed likely to occur at a time of strong time derivative of the concentration. Figure 9 shows the resulting description of phenology for the surface chlorophyll concentration, as obtained for the prior

ensemble (top panels), the LETKF analysis (middle panels) and the MCMC sampler analysis (bottom panels). The figure displays quantiles of each of these ensembles (from left to right: 20%, 50% and 80%). It is important to emphasize here that phenology has been computed first for each ensemble member, and then the quantiles, not the other way around. The result is a probability distribution for phenology, which we describe by a few quantiles. In the figure, we can see that the prior uncertainty in phenology is quite large, and that this uncertainty is strongly reduced by the observation constraint, in a way that is very

similar in the two methods.

Furthermore, from the ensemble experiments, it is possible to explore if the phenology of chlorophyll is linked to the phenology of zooplankton (using the same definition as above). Fig. 10 shows a scatterplot of these two dates at the same location already used for Fig. 4 ($36°W$, $44.25°N$, surface value, at the center of the region). The figure compares the result obtained with the LETKF algorithm (left panel) and the MCMC sampler (right panel). The figure displays members of the

prior ensemble (black dots) and of the ensemble analysis (blue dots). The small black dots in the right panel represent members of an augmented version of the prior ensemble (200 members), with covariance localization. It is obtained with the MCMC sampler, used exactly as for the analysis (at a much lesser cost), but without the observation constraint. It is more representative of the prior ensemble that is actually used by the MCMC sampler, since it includes localization.

In the figure, we can see that the prior ensemble mainly opens three possible time windows in which chlorophyll phenology

can take place. They are represented in the figure by the three light green areas, which can be seen to correspond to peaks of the prior surface chlorophyll concentration in Fig. 4. These three windows can presumably be associated to favourable conditions offered by the physical forcing (in terms of light, temperature, and/or mixing) for the phytoplankton bloom to occur. The specific phenology in each member then depends on how the biogeochemical model behaves, as a function of the stochastic perturbation applied to each of them. From these three modes of the prior distribution, the observation constraint makes the

ensemble analysis select just the third one, in which the posterior probability concentrates (similarly with the two methods). Correspondingly, the phenology of zooplankton displays the same three windows in the prior ensemble (in light blue in the figure), with a small general shift forward in time. But the posterior uncertainty is here larger: $\pm 5$ days for zooplankton rather than $\pm 1$ day for chlorophyll (at this location).

About the posterior uncertainty in the phenology of zooplankton, it must be emphasized that this largely depends on the

assumptions that have been made to describe the prior uncertainties. For instance, in the prior ensemble simulation described in section 2, uncertainties in the grazing of phytoplankton by zooplanktoon have not been taken into account. These uncertainties





**Figure 9.** Quantiles of phenology (in days). Prior ensemble (top), LETKF analysis (middle) and MCMC sampler analysis (bottom).





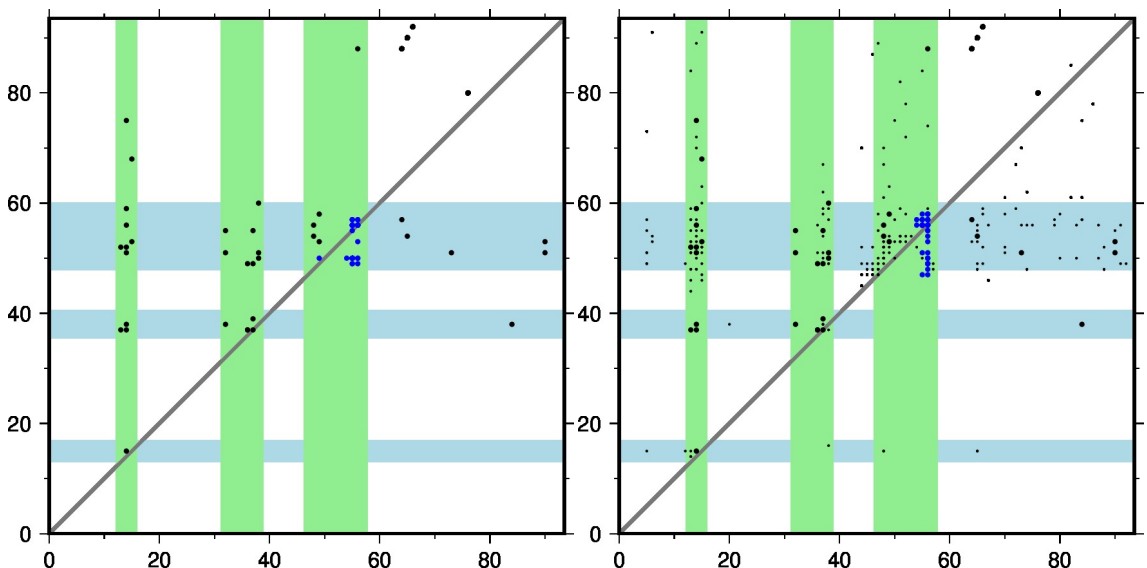

**Figure 10.** Scatterplot of phenology (in days) for chlorophyll (x-axis) and zooplankton (y-axis). At $36°W$, $44.25°N$. With the LETKF algorithm (left) and the MCMC sampler (right). Prior ensemble (black), augmented prior (small black dots), analysis (blue).

may have an influence on the dependence between the behaviours of phytoplankton and zooplanktoon, and thus increase uncertainties in the phenology of zooplankton, as compared to what is shown in Fig 10.

### 5.3.2 Trophic efficiency.

Trophic transfer efficiency (or trophic efficiency) measures the part of energy that is transferred from one trophic level to the next and is a common indicator used to characterize food availability to high-trophic level organisms of economic or ecological importance and, for instance, how this availability is affected by environmental changes (Eddy et al., 2021). Trophic efficiency is usually computed as the ratio of production at one trophic level to production at the next lower trophic level. However, since those specific diagnostics are rarely recorded explicitely, it is common to use the ratio between the biomass of upper and lower

trophic levels as a proxy measure of the trophic transfer efficiency within the food web (Armengol et al., 2018; Eddy et al., 2021). Here, trophic efficiency is evaluated as the ratio between the biomasses of primary producers and consumers vertically integrated over the 0–200m layer.

The computation of this indicator first requires vertical profiles of phytoplankton and zooplankton, and this depends on the ability of the inverse methods to extrapolate the surface information in depth. Fig. 11 displays vertical profiles of chlorophyll

concentration (left panels) and microzooplankton concentration (right panels), as obtained with the LETKF algorithm (top panels) and the MCMC sampler (bottom panels). It is shown for the same location as in Fig. 10 ($36°W$, $44.25°N$, at the center of the region) and for the median date obtained for the chlorophyll phenology (day 55, May 8) in the ensemble analyses. By





virtue of the anamorphosis transformation, all posterior ensemble members gently follow the same general vertical structure as the prior ensemble, without overshootings or obvious inconsistent behaviours. Only the spread of the ensemble is reduced, towards values that are somewhat different in the two methods. At that date and location, the chlorophyll concentration is larger with the MCMC sampler, and the microzooplankton concentration is smaller.


**Figure 11.** Vertical profile of chlorophyll (in mg/m$^3$, left) and microzooplankton (in mmol/m$^3$, right). At 36°W, 44.25°N. May 8, 2019. With the LETKF algorithm (top) and the MCMC sampler (bottom). Prior ensemble (black), analysis (blue).



More importantly, we see that, in all profiles, the depth of the active layer is about the same in all members, which means that we assumed no prior uncertainty on this. This can only mean that uncertainties are missing in the prior NEMO/PISCES simulator (presumably in the physical component), and the consequence is that our results probably underestimate uncertainties in the vertical integrals. In any case, these uncertainties could not have been controlled using ocean colour observations only, and would have required involving other types of observations.

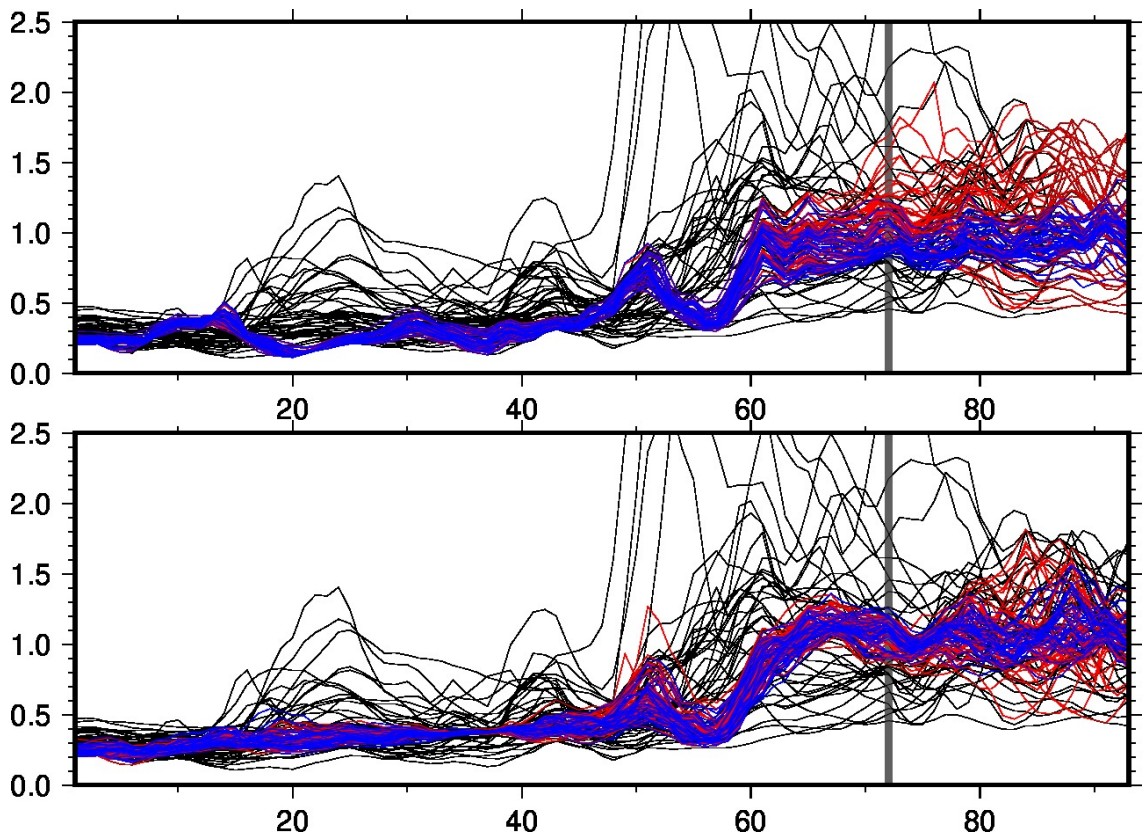

**Figure 12.** Timeseries of trophic efficiency. At $36°$W, $44.25°$N. With the LETKF algorithm (top) and the MCMC sampler (bottom). Prior (black), analysis (blue), forecast (red).

With this caution in mind, Fig. 12 displays time series of the vertically integrated trophic efficiency at the same location, as obtained with the LETKF algorithm (top panel) and the MCMC sampler (bottom panel). The figure is organized exactly as Fig. 4, at the same location, but for a different variable. From day 0 to day 60, the ratio slightly increases from 0.3 to 0.5, a range that is in line with the value given in this area by the trophic model of Negrete-García et al. (2022). After the bloom (see Fig. 4), the trophic efficiency increases towards values closer to 1, and above 1, which is indicative of a transition from a bottom-up to a top-down control of the phytoplanktonic population. In the prior ensemble, we can still distinguish the 3 possible windows for the occurence of the bloom, while it is reduced to 1 in the posterior ensemble (with a smaller spread),




consistently with the posterior phenology estimates. However, the quantitative accuracy and reliability of the trophic efficiency estimates entirely depend on the reliability of the prior ensemble, and in particular on the reliability of the correlation between the trophic efficiency and the observed variable (see discussion at the end of section 5.3.3).

### 5.3.3 Downward flux of particulate organic matter.

The downward flux of particulate organic matter relates to the biological gravitational pump, i.e. the sinking of organic matter under direct action of gravity. It is one constituent of the biological carbon pump, which includes the ensemble of processes that transfer carbon from the surface ocean (where it is balanced with atmospheric $CO_2$ concentration through air-sea exchanges) and deeper oceanic layers where carbon is considered as 'stored' on larger time scales (Claustre et al., 2021). There are many ongoing discussions on how to optimally infer biological carbon pump indicators from model simulations, sustained by the obvious importance of such indicators in the frame of global change (e.g., Galí et al., 2022). In our case, we stick to the simple definition of multiplying the concentrations of two classes of organic matter at the 100m horizon (below the active layer, see Fig 11) with their specific sinking velocities, and summing up both contributions. To understand the results below, it is important to note that: (i) the model simply considers fixed sinking rates (parameters) for each class of POM, and (ii) those parameters are not part of the uncertainty considered to build the prior ensemble. The spread of this indicator in the prior ensemble remains therefore intimately related to that of phytoplankton production.

Fig. 13 displays a scatterplot of this indicator (y-axis) vs surface chlorophyll concentration (x-axis) for May 26, 2019, at the same location as before (36°W, 44.25°N, at the center of the region), as obtained with the LETKF algorithm (left panel) and the MCMC sampler (right panel). The figure displays members of the prior ensemble (black dots), of the ensemble analysis (blue dots), and of the 4-day ensemble forecast (red dots). With both methods, the spread of the analysis and forecast is strongly reduced as compared to the prior ensemble. The two methods produce a similar ensemble analysis. With the MCMC sampler, the forecast is compatible with the analysis, but with a larger spread. With the LETKF algorithm, however, the 4-day forecast is incompatible with the analysis. This happens to be a location at which the 4-day LETKF forecast (but not the 1-day forecast) is biased as compared to the analysis. This could already have been observed in Fig. 7, where, at the center of the region, there is a pattern with a low value of the surface chlorophyll concentration in all quantiles, which is incompatible with both the analysis and the observations. This bias in the forecast here translates on the forecast of the indicator.

The reliability of the indicator estimate depends on the reliability of the prior correlations between this indicator and the observed variable (as described by the black dots in the figure). The reliability of the forecast also depends on the reliability of the time correlation (and the time localization scheme). Both depends on the modelling assumptions embedded in the NEMO/PISCES simulator. These assumptions here combines a deterministic framework (providing information about the behaviour of the system) and a stochastic parameterization (accounting for uncertainties). As in any modelling system, unreliable assumptions will produce unreliable results. For instance, in this example, we neglected uncertainties in the sinking velocities, so that the confidence in the indicator is certainly overestimated. It is even possible that no valuable information about this indicator can be obtained from surface chlorophyll only, in view of the current modelling knowledge and missing uncertainties.





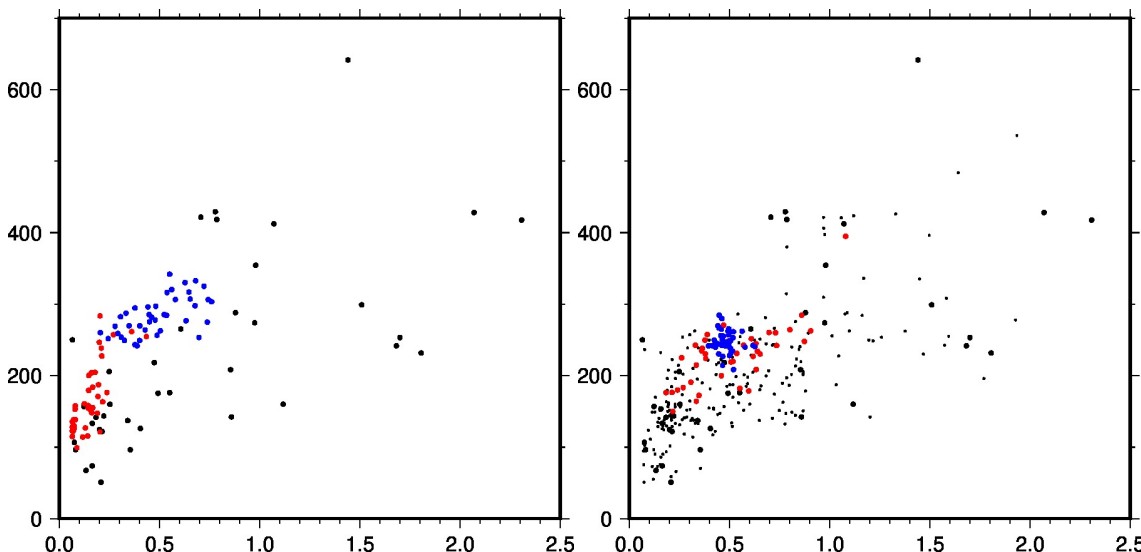

**Figure 13.** Scatterplot of surface chlorophyll concentration vs POC flux at 100 m depth. At 36°W, 44.25°N, on May 26, 2019. With the LETKF algorithm (left) and the MCMC sampler (right). Prior (black), augmented prior (small black dots), analysis (blue), 4-day forecast (red).

But the same difficulty can be expected for any complex system, in which the confidence in the assumptions is bound to be low at the beginning, as long as little information is available, and then progressively enhanced.

## 6    Conclusions

In this paper, a simplified approach has been introduced to perform a 4D ensemble analysis describing the evolution of the ocean ecosystem. In our example, it is based on prior ensemble statistics from a stochastic NEMO/PISCES simulator and ocean colour observations. The observations are used as constraints to condition the 4D prior probability distribution. The results show that it is possible to produce an ensemble solution that is continuous in time, consistent with the observations, and with a reliable description of the posterior uncertainty (at least for the observed variable). Furthermore, attempts have

been made to extrapolate the results into the future from past observations. The resulting 4D ensemble statistical forecast is shown to be most often reliable (at least for the observed variable), and to contain valuable information about the evolution of the ecosystem for a few days after the last observation. However, as a result of the short decorrelation time scale in the prior ensemble, the spread of the ensemble forecast is increasing quickly with time.

In terms of method, the inverse problem has been solved using two very different algorithms: the analysis step of the LETKF

(with domain localization) and an MCMC sampler (with covariance localization). The performance of the two algorithms are enhanced by applying a nonlinear transformation to the estimation variables (anamorphosis). These two methods being so different from each other, the most important conclusion that can be drawn is certainly that common features displayed by the



two results validate each other. Differences between the results are more difficult to interpret, because it is difficult to separate the effect of the method and the effect of the tuning of localization (which cannot be made equivalent in the two algorithms). On the one hand, covariance localization (with a localizing correlation function) is more flexible, and it has a better theoretical justification, so that the Bayesian inverse problem remains global, and the local correlation structure of the prior ensemble can be preserved better. On the other hand, domain localization (with decreasing influence function) is more like an ad hoc solution with only a few parameters, and there is no global Bayesian formulation of the problem anymore, so that there is more risk of triggering non-physical small scales when the local domain are shifted from each other (to solve the global problem piece by piece). However, in our experiments, it can happen that this simple scheme, with just a weighted decrease of the influence of the past observations, can produce a more accurate ensemble forecast, but also occasionally bigger biases.

The key practical advantage of the approach presented in this study, as compared to standard data assimilation, is to decouple the resolution of the inverse problem from the application of the complex numerical model (here the NEMO/PISCES stochastic simulator), which is only used as a supplier of prior input data. This has given us the possibility to focus on a small subregion, in which the inverse problem is well-conditioned, and to solve the problem globally in time for a full 3 months time period, thus avoiding the filtering approach, in which only past observations can influence the analysis at a given time. Without the need to reinitialize the model, it has also been possible to concentrate on a few variables of interest, and indicators that are not model state variables. As long as the prior ensemble statistics are reliable for a subset of the model variables, an analysis and forecast can directly be obtained for these variables and associated indicators without going through the burden of estimating the full state vector. In terms of method, this simplified approach has allowed us to try using a more general iterative algorithm to solve the problem, thus avoiding the Kalman approach, in which only linear constraints are possible (linear observation operator and Gaussian observation errors). In our results, this did not bring much improvements because much care has been taken to perform the anamorphosis transformation of the non-Gaussian observation error probability distribution required by the Kalman algorithm. But this is still an approximation, which is not necessary in the MCMC sampler. Moving to a nonlinear method, solving the problem globally, also offers new perspectives in terms of constraints that can be applied to the solution, such as nonlocal and nonlinear observation operators, or nonlocal and nonlinear dynamical constraints (as long as the associated cost function is not too expensive to evaluate).

The main theoretical shortcoming of this approach is that the complex dynamical model is no more directly used to constrain the solution. However, it must be recognized that this is not always possible in practice. As long as most of the model state variables remain unobserved (and there are 23 state variables in PISCES) and often poorly correlated to the observed variables, it can be a very difficult task to initialize properly a dynamical model forecast (or to adjust an even larger number of model parameters), especially if the model is very sensitive to small inconsistencies in the initial state. Ad hoc adjustments and simplifications are then usually needed to avoid model failure or unrealistic results. In other words, in case of non-observability and non-controllability of the dynamical system, the problem can become ill-conditioned, and it can be worthwhile to start from a simplified approach in which these problems do not occur.

Regarding the operational applications, this approach offers the possibility to be more flexible and less dependent on the behaviour of the global data assimilation system. It can quickly focus on a specific region of interest and produce targetted



products to meet dedicated users' requirements, at small additional cost. This can be done without any new technical devel-
opement, using the same algorithm already in use in the dynamical data assimilation system (e.g. the LETKF algorithm), and
the simplified framework can serve as a testbed for more advanced inverse methods (e.g. the MCMC sampler).

The main condition to perform such 4D analysis and forecast of the evolution of the ecosystem is to produce a *reliable* prior
ensemble simulation based on the complex dynamical model. This is not an easy prerequiste, which requires identifying the
most important sources of uncertainty in the system, and developing stochastic perturbations schemes to take them into account.
Reliability must then be checked against observations, so that the inversion can concentrate on regions where uncertainties
have been sufficiently understood and where reliable products can be delivered. Concerning the non-observed variables (like
the ecosystem indicators), the reliability of the results entirely depends on the assumption made to produce the prior ensemble,
which must be questioned and checked wherever possible. In our results about the indicators, we have mentioned that important
uncertainties have been neglected, so that the posterior uncertainty on the indicators is certainly underestimated. However, the
situation is similar in dynamical forecasting systems, in which the reliability of the ensemble forecast also depends on the
modelling assumptions. Any progress towards a better understanding of the model uncertainties is thus directly beneficial to
both types of system.

*Code availability.* The ensemble simulations have been performed with NEMO4.0, including stochastic parameterizations, as provided
through the SEAMLESS project (doi:10.5281/zenodo.6303007). The ensemble observational update (with the LETKF algorithm and the
MCMC sampler) and the computation of the probabilistic scores have been performed using the EnsDAM (github.com/brankart/ensdam) and
SeSAM softwares (github.com/brankart/sesam), using a set of shell scripts to perform the various operations (github.com/brankart/ensemble-
ocean-colour).

## Appendix A:  Anamorphosis transformation of the observations

The following algorithms have been implemented to address the problem of transforming the observations. We first consider a
simplified particular case allowing a more efficient algorithm, before addressing the general problem. In both case, the objective
is to produce an unbiased transformed observation, with a consistent observation error standard deviation.

### A1   The observation error pdf is symmetric and does not dependent on the true state

In this particular case, the transformation of the probability distribution for observation error is straightforward, and can be
obtained using the following algorithm for each observation (assuming independent observation errors):

1. Compute the anamorphosis transformation $\mathcal{A}$ for each observed quantity from the ensemble equivalent of the observa-
tion: $\mathcal{H}(\mathbf{x}_i), i = 1, \ldots, p$ (where $\mathbf{x}_i$ is an ensemble member and $\mathcal{H}$ is the observation operator).

2. Produce a sample of perturbed observations: $y_j^o = y^o + \epsilon_j$, where $y^o$ is the observation and the perturbations $\epsilon_j$ are
     sampled from the observation error probability distribution.



3. Transform the sample using the anamorphosis transformation $\mathcal{A}$.

4. Use the mean and covariance of the transformed sample as parameters for the transformed observation error probability distribution (assumed Gaussian).

## A2  General observation error probability distribution

In the general case, when the observation error probability distribution depends on the true state of the system or when it is not symmetric, perturbations with this distribution cannot be added to the observations. They can only be applied to a model equivalent to the observations $\mathcal{H}(\mathbf{x}_i)$. In this case, applying the simplified algorithm above can lead to substantial biases in the transformed observations, especially for bounded variables (when observations are close to the bounds). It is then important to use a more general algorithm to obtain the transformed observation and observation error:

1. Sample a rank $r$ for the observation error (uniformly between 0 and 1).

2. Perturb each member of the ensemble with an observation error with this same given rank: $y_i = F^{-1}(r)$, where $F$ is the cumulative distribution function (cdf) of the observation error probability distribution $p[y_i|\mathcal{H}(\mathbf{x}_i)]$, conditioned on the member that is being perturbed.

3. Compute the anamorphosis transformation $\mathcal{A}$ from this transformed ensemble.

4. Transform the observation with $\mathcal{A}$ to obtain a transformed perturbed observation.

5. Repeat the above steps for a sample of ranks to obtain a sample of transformed observations.

6. Use the mean and covariance of the transformed sample as parameters for the transformed observation error probability distribution (assumed Gaussian).

It is easy to see that this more general algorithm is equivalent to the simplified algorithm above in the particular case of observation errors that have a symmetric probability distribution, which does not depend on the state of the system.

*Author contributions.* MP performed the ensemble experiments. JMB proposed the approach, contributed to the production of the results and initiated the writing of the mansucript. AC contributed to the definition, computation and interpretation of the ecosystem indicators. EC provided expertise on data assimilation methods. PB initiated the definition of the problem, as a contribution to the SEAMLESS project and contributed to the definition of the approach. All contributed to the design of the experiments and the writing of the manuscript.

*Competing interests.* None of the authors have any competing interest.



*Acknowledgements.* This work was funded by the SEAMLESS ("Services based on Ecosystem data AssiMiLation: Essential Science and Solutions") project which has received funding from the European Union's Horizon 2020 research and innovation programme under grant

agreement No 101004032. The calculations were performed using HPC resources from GENCI-IDRIS (grant A10-011279). The perceptually uniform colour-maps 'tofino' and 'roma' (developed by F. Crameri, doi: 10.5281/zenodo.2649252) is used in this study to prevent visual distortion of the data.



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
