# Peer review of "Ensemble analysis and forecast of ecosystem indicators in the North Atlantic using ocean colour observations and prior statistics from a stochastic NEMO/PISCES simulator"

_EGUsphere, 2023_

## Author Comment (AC1)

**Answers to the reviewer's comments**

We thank the Reviewer for his/her careful reading of our paper, and for his/her remarks that will help improve the clarity of the manuscript. We did our best to take them into account as explained below. Our replies and comments are given in normal type, while the original comments from the Reviewer are in bold/italics.

*The authors present ensemble-based statistical experiments, akin to data assimilation, in which a static ensemble provides the prior statistics, and in which the posterior statistics are computed for only a part of the full model state. The study is interesting and presents some carefully worded, compelling results. The manuscript is easy to follow and mostly well written, and I have only a few general comments.*

**# general comments**

*The use of an offline or static ensemble to generate various statistics is an interesting one, but maybe not as novel as the manuscript suggests. Line 66 states that "For these reasons, there are certainly practical situations in which it would be interesting to append such a 4D statistical analysis and forecast to existing ensemble data assimilation systems. They may serve as a baseline to compare with the dynamical ensemble forecast, and as a possible substitute whenever useful." Such an approach has been implemented before, and is often referred to as ensemble optimal interpolation (EnOI). Evensen presents the idea as a computationally cheaper alternative to the EnKF (Evensen, 2003). More recent implementations are, for example, Oke et al. (2010) and Mattern and Edwards (2023) which is using it for data assimilation with ocean color observations. They all rely on performing data assimilation with a static ensemble based on existing model output.*

*The manuscript is mostly well written, but sometimes assumes too much knowledge about the methodology from the readers. A few more sentences in the right places would be helpful to readers who are not familiar with Brankart (2019) and the few other papers that this manuscript's methodology is based on. I have highlighted specific instances in my comments below.*

*The figures in this manuscript are very useful, but they are missing labels, units and information. Generally, axis and color bar labels are missing. Further adding legends (e.g., "prior", "analysis") and labels (e.g., "MCMC" or "LETKF") would let many readers get most of the information without having to study the caption. Again, I have highlighted specific instances in my comments below, but all figures require labels.*

*Evensen G., 2003: The Ensemble Kalman Filter: theoretical formulation and practical implementation. https://doi.org/10.1007/s10236-003-0036-9*

*Oke P.R., Brassington G.B., Griffin D.A., Schiller A., 2010: Ocean data assimilation: A case for ensemble optimal interpolation. https://doi.org/10.22499/2.5901.008*

*Mattern J.P., Edwards C.A., 2023: Ensemble optimal interpolation for adjoint-free biogeochemical data assimilation. https://doi.org/10.1371/journal.pone.0291039*

Yes, we fully agree that there is no new data assimilation method developed in this paper, which entirely relies on existing techniques. The difference is only in how these methods are applied to solve the problem.

Our approach can indeed be viewed as a direct 4D application of an ensemble OI algorithm over a given time window. Usually, however, in standard ensemble OI (as in the papers given by the reviewers), these time windows are kept quite short (typically a few days) and there is still an alternance between such 3D (and sometimes 4D) ensemble analyses, and a dynamical model forecast (which requires re-initializing the model from the analyses). Moreover, ensemble OI is usually defined by the use of a prescribed prior ensemble, which is derived from historical data and which is kept the same for all assimilation cycles (or with a prescribed seasonal variation).

In our approach, a further simplification of the problem is applied, in which there is no more cycling between the 3D or 4D ensemble updates and the dynamical model (and thus no more initialization of the model from the analyses). The ensemble solution is computed at once over an extended time window (several months, which required including a time localization algorithm), and the prior ensemble is computed from the dynamical model running over the same time window, so that there is no assumption of stationarity of the statistics, which is usual in classic ensemble OI (see also answer to Reviewer 2 about this). The fact that this can be achieved with existing methods and tools without any change is presented in the paper as a strength of the approach.

The following text (with the references provided by the reviewer) has been added in the introduction to further clarify this point, and make a more explicit link with ensemble OI :

« In practice, this can for instance be achieved by a direct application of just one analysis step of the ensemble optimal interpolation (EnOI) algorithm (e.g. Evensen, 2003 ; Oke et al., 2010), for a 4D estimation vector (thus embedding the time evolution of the system, as in Mattern and Edwards, 2023), but over an extended time window. However, it is important to emphasize that, unlike EnOI, we do not use historical data to prescribe the prior statistics, but an ensemble simulation that is specifically performed for the requested time period. This is necessary if we want to avoid assuming stationarity of the statistics. »

**# specific comments**

**L 21: "thus somehow validating each other.": The "somehow" makes it sound too casual, I would suggest something like: "thus providing evidence for each other's estimates."**

Yes, we changed the formulation into « thus providing support to each other's estimates ».

**L 47: "to make an additional use of these expensive data": While I know what is meant here, "data" was previously used to refer to observational data. I would suggest adding "model" to make it clear that this is referring to model output. Furthermore, "obtained from a prior ensemble model simulation" could become "obtained from prior (ensemble) model simulations".**

Done.

**L 148: "multiplicative noise in the metrics of the model grid": What exactly does this mean, please explain in a bit more detail.**

This means that the uncertainty in the location of the fluid parcels is indirectly simulated by introducing random perturbations to the model grid, and more specifically to the horizontal size (dx, dy) of the grid cells. We have tried to be more explicit about this in the paper, but it is difficult to tell the full story in a few lines. More explanation is given in Leroux et al. (2022), which is freely available in Ocean Science.

**L 151: "two-dimensional maps of autoregressive processes": Are these horizontal maps that are smoothed in some way? A bit more information could be useful to the reader.**

Yes, they are smoothed to introduce spatial correlation in the different sources of noise. The correlation scales are given in Table 1. The caption of the table now includes the addtional sentence : « The horizontal correlation is obtained by applying a smoothing operator. »

**Figure 1, 2 and others: It would be helpful to add the property and units to the color bars of each figure (especially in Fig. 2, where they are not mentioned in the caption). Also, it would be useful to mention the date.**

Yes, thanks. This has been fixed.

**L 279: "..., thus ensemble members.": This statement is difficult to understand, please rephrase.**

This has been rephrased.

**L 288: Why not mention right away (in the first or second sentence of this paragraph) that this implementation is based on Brankart (2019). This comment may also apply to the previous paragraph if the LETKF implementation is very similar to that in Brankart et al. (2003) which is mentioned only at the end of that paragraph.**

We think that this would give the false impression that these methods have been introduced in these papers. Actually, what we have done in these papers are just variants or specific implementations of pre-existing methods.This is why we prefer giving earlier references at the beginning of the paragraph, and then reference to our specific version of the method at the end (even if this indeed make the presentation more intricate).

**L 295: "zero correlations are approximated by non-zero correlation": Perhaps generalize this statement a bit b/y saying that low correlations are typically overestimated with a small ensemble?**

Yes, indeed. This has been changed.

**L 315: "... (with a normalized Gaussian random factor).": What is a normalized Gaussian random factor? Is it multiplied with the vector? This is already a long sentence, I suggest dividing it into two sentences and providing a bit more context.**

Yes, indeed, this parenthesis was unclear. We have removed the parenthesis and added the sentence :
« This provides the time-space pattern of the perturbation, which is then multiplied by a Gaussian random factor (with a standard deviation equal to 1). »

***Figure 4: It would be useful for the reader to add labels to the x- and y-axes. Furthermore, turning the x labels into dates, adding "LETKF" and "MCMC" into the top-left corner of the respective panel, and a legend for "prior", "analysis" and "forecast" would also make the plots much more accessible.***

Done.

***L 385: "remote observation is missed, ..." → "remote observations are missed" (or ignored)***

Done.

***L 410: I would suggest moving this paragraph to the next section, following the introduction of the CRPS score.***

Done.

***L 418: Mention that CRPS stands for Continuous Ranked Probability Score when it is first used, and provide a reference for it.***

Done.

***Figure 7 and others: additional panels with the inter-quantile range (e.g. 80%ile - 20%ile) could be useful to better visualize differences in the ensemble spread.***

Yes, maybe. But this would also dilute the information, and lead the user to go through even more figures, which are already numerous.

***L 542: Out of curiosity, how bad would be the use of a log-transformation (plus perhaps a normalization) to perform the anamorphosis?***

We did not try this option. It is thus difficult to know. Presumably, this would not make a big difference for the observed variable (surface cholorophyll). But the extrapolation to non-observed variables would be more sensitive, since the correlations between the transformed variables are not the same. Besides, there are variables (like trophic efficiency for instance) for which the log-transformation is not appropriate.

***L 560: "the first date at which the chlorophyll concentration reaches half of its maximum value over the whole time period": half of the maximum value at that particular location or half of the maximum value in the domain?***

At that particular location. This has been clarified.

***L 570: What is the phenology in the observations, do the LETKF and MCMC solutions get closer to the observed values?***

There is no observed value of phenology. It cannot be directly diagnosed from L3 observations because of the many time gaps. One of the outcome of this paper is actually to fill these gaps (with an ensemble description of the resulting uncertainty) and be able to diagnose phenology from the result (with its own ensemble description of uncertainty). We could have compared to L4 chlorophyll products, which are obtained by time and space interpolation of the L3 products, but this cannot be considered as an observation reference. There is no reason to believe that this simple interpolation (without uncertainty estimates) would be more reliable or more accurate than the one performed in this study.

***L 645: "This happens to be a location at which the 4-day LETKF forecast (...) is biased ...": Mention that this is the chlorophyll forecast (I presume) explicitly.***

Done.

***L 657: "But the same difficulty can be expected for any complex system, in which the confidence in the assumptions is bound to be low at the beginning, as long as little information is available, and then progressively enhanced." I don't quite understand this sentence, what is being enhanced here? I entirely agree with the point that in complex systems, sources of uncertainty are often ignored or not modeled adequately, and that can lead to artificially low uncertainty estimates in certain indicators.***

Yes, this sentence was probably too general and did not bring much useful information. It has been changed according to the suggestion made by the reviewer :

« But the same difficulty can be expected for any complex system, in which uncertainties are often ignored or not modeled adequately, which can lead to artificially low uncertainty estimates in certain indicators. »

***L 698: "The main theoretical shortcoming of this approach is that the complex dynamical model is no more directly used to constrain the solution." But doesn't the dynamical model provide the prior ensemble which does affect/constrain the posterior estimates?***

Yes, indeed, the dynamical model is still indirectly used to constrain the solution through the statistics of the prior ensemble. We argue in the paper that, if these statistics can be used adequately, this is already sufficient to obtain a useful solution to the problem, especially if it is difficult to estimate an initial condition that is consistent enough to produce a sensible forecast with the dynamical model. However, what we can provide with this approach is only a statistical forecast, which is thus not a solution of the dynamical model. It would be a legitimate critics to consider that this is a shortcoming of our approach, especially in practical situations (e.g. enough observations), in which it is possible to produce a sensible initial condition for the dynamical model.

---

## Author Comment (AC2)

**Answers to the reviewer's comments**

We thank the Reviewer for his/her careful reading of our paper, and for his/her remarks that will help improve the clarity of the manuscript. We did our best to take them into account as explained below. Our replies and comments are given in normal type, while the original comments from the Reviewer are in bold/italics.

*This study proposes a methodology for combining numerical model output with observations. The framework is that of Bayesian analysis. In contrast to most data assimilation, however, the emphasis is on using offline pre-generated numerical model ensembles as the prior information, rather than embedding a numerical model in the analysis scheme. The application is for multiple biogeochemical variables (while observing only one). I played around with a similar approach years ago (using an enKF, but with explicit time evolution), but never did a real application nor proper assessment (we decided to take another approach for our analysis of ocean carbonate variable and so it ended there). Hence, I like the approach, and am pleased that someone has taken it forward to the community, as I always thought it would be useful. My opinion is that there is a real need for these types of space-time analysis procedures that don't require explicit running of numerical models within the estimation procedure (but still use numerical model information). I note that lots of statistics people are doing problems like this with sophisticated spatio-temporal approaches (e.g. integrated nested Laplace approximations, INLA), but these tend not to be very accessible the ocean community. The approach taken here is straightforward, builds on basic data assimilation principles, gets decent results, and should be accessible to most ocean data analysts. Hence I recommend publication with some minor revisions.*

*COMMENTS*

*There is a strong link with the foundational approach of optimal interpolation (which parallels the Kalman filter observation step). Since most people know about OI, it might be useful to make a quick note of it in order to make the approach more clear to the non-expert reader.*

Yes, we agree that a link with OI was missing in the paper. This has been added in the introduction. Sea also our answer to reviewer 1 about this.

*My main confusion in understanding the methodological development was how time was incorporated into the analysis. After a couple re-reads, I see this is made clear early on when you define the state vector (its dimension includes time). But this could be brought out more explicitly. When most people see that you have used the Kalman filter update machinery, they will wonder about evolution through time. Part of my confusion may also have arisen since when I did my version of this problem, I actually ran it sequentially in time with a daily time step, and did the (spatial) observation update whenever measurements were available. My time correlation model was an auto-regressive one, and I used an enKF/enKS methdology. Your time correlation is implicitly embedded in the space-time covariance matrix that defines the multivariate state.*

Thank you. It is indeed very important that this point can be understood clearly. We have added the following sentences in the paper to clarify this.

In the introduction, while making the link with ensemble OI : « In practice, this can for instance be achieved by a direct application of just one analysis step of the ensemble optimal interpolation (EnOI) algorithm (e.g. Evensen, 2003 ; Oke et al., 2010), for a 4D estimation vector (thus embedding the time evolution of the system, as in Mattern and Edwards, 2023), but over an extended time window. »

In section 3.1, defining the 4D estimation vector : « As compared to classic sequential data assimilation (like the ensemble Kalman filter), the difference here is that the whole 3-month time sequence is packed together in the 4D estimation vector. This is makes the problem bigger, but also allows us to concentrate on a small subregion and a few selected variables. »

*Lines 140-150. I found this discussion of uncertainties 2 and 3 confusing. I get that you are trying to account for unresolved scales. There is likely a better plain language way of saying what you are doing.*

Yes, we have tried to clarify this by adding a short simple explanation on what we are doing, in one sentence at the beginning of each of these paragraphs, before going to the technical explanation.

Uncertainty 2 : « In the biogeochemical model, another important source of uncertainty is the effect of the unresolved scales on the large-scale component of the biogeochemical tracers. As a result of the nonlienar formulation of the model equations,... »

Uncertainty 3 : « Unresolved scales in the physical component of the model also produces large-scale

effects that are difficult to parameterize, and thus produce uncertainties that are not easy to simulate. »

***Do you think it is proper to equate a 4D inverse problem with a Bayesian estimation? I know there are links, but you have to hand-wave a lot to explain them. Why not just say you used a Bayesian method?***

We just say that we use the Bayes theorem to solve the problem. We have a prior probability distribution describing the vector of variables to be estimated, and then conditions that we apply on this prior distribution (in our case, observations). Then, we make approximations on the shape of these distributions to solve the problem (which are different in the two methods that we have applied). How close we are to an exact Bayesian estimation will depend on the validity of these assumptions. For instance, there are much less assumptions in the MCMC sampler, since the observation condition can be applied using the native probability distribution for the observation uncertainties (i.e. without approximations on the likelihood function).

***Nice job on highlighting the difficulties of using small ensembles, partial observation of the state, and the need to estimate a big multivariate state. The tricks to make this work (like localization) are appreciated by the reader. Similarly, nice job on trying out some "ecological indicators" which emphasize the multivariate state and how measurement on one variable can tell you about other variables (and project to depth). However, the ecological indicators are to me not so central, and if shortening the paper was required I would omit these.***

Thank you. Concerning the estimation of indicators, we think that it is useful because it shows how the method can be used to obtain a direct estimation of variables of interest, and why a model-derived ensemble is needed (see answer to comments below).

***The central quantity in such an estimation problem is the ratio of observation variance to the model (ensemble) variance. This will dictate how far the prior is moved by the observations in creating the posterior. This is captured in your Probabilistic Scores, I think. But with simple messaging, the point could be made clearer.***

Yes, it is important to quantify how much information is brought by the observations, which depends on the ratio between the observation error variance and the background error variance. In our problem, however, the variables are non-Gaussian, so that we decided to use probabilistic scores rather than variances. We added the following sentence in the paper to clarify this point.

« In the case of Gaussian variables, the gain of information brought by the observations (i.e. the resolution component of the score) is often characterized by variance ratios, which could have been computed here for the transformed variables (by anamorphosis). But we have preferred providing an assessment of the original concentration variable using the CRPS score. »

***Figures need more details. You don't label the axes in some. You don't define what variables is being plotted in others. Etc.***

Yes, thanks. This has been fixed.

***An alternative approach is to use a parametric space-time covariance matrix. For example, a common approach is to use a Matern covariances for space, and auto-regression in time (and generally assume space-time separability for simplicity). This contrasts to your sample covariance matrix with post-processing (localization). Thoughts? Pros and cons?***

Yes, in the optimal interpolation algorithm, it is possible to use parametric space-time correlation structures. However, it is much more difficult to specify cross-variable correlation structures with these methods, and thus to provide estimates of non-observed variables. This is why model-derived ensemble covariance structures are used. This point is specifically illustrated by the direct estimation of ecosystem indicators from the observations using (space and time dependent) model-derived correlation structures.

***The way you do MCMC would be likely be called approximate Bayesian computation. That is, you make use of a cost function, rather than a more exact likelihood ratio.***

In the MCMC sampler, the observation cost function is just the log of the likelihood ratio (as explained in the text following equation 2). It can be evaluated exactly, without the usual approximation of a quadratic cost function. The resulting sample is thus a sample of the posterior distribution, without approximation in the observation condition. However, the prior distribution is still assumed Gaussian (for the transformed variables, after anamorphosis), as part of our approximate formulation of the Bayesian estimation problem.

***Stationarity is, in general, likely the assumption that limits the forecast horizon. Training on one time period, and applying it to another time period is predicated on stationarity. However, your short term forecasts of a few days means this is not an issue since it is driven by de-correlation timescales.***

Yes, this is a very important point. In our approach, there is no assumption of stationarity of the statistics. Stationarity of the statistics is indeed usually assumed in classic optimal interpolation or in learning methods to transpose statistics obtained from past data to the present situation. In our approach, the prior ensemble is obtained from an ensemble model simulation that is specifically performed for the time period where the estimation is requested. This is especially important when the model is constrained by a time-dependent forcing function (in our case, the atmospheric forcing), which makes the ensemble statistics very instationary (maybe mostly seasonal, but not only). The following sentences have been added in the introduction to insist on this point.

« However, it is important to emphasize that, unlike EnOI, we do not use historical data to prescribe the prior statistics, but an ensemble simulation that is specifically performed for the requested time period. This is necessary if we want to avoid assuming stationarity of the statistics. »